# Automating General Movements Assessment with quantitative deep learning to facilitate early screening of cerebral palsy

Qiang Gao [1,6], Siqiong Yao[1,2,6], Yuan Tian[3], Chuncao Zhang[3], Tingting Zhao[4], Dan Wu[4], Guangjun Yu[4,5] ✉ & Hui Lu [1,2,4] ✉

The Prechtl General Movements Assessment (GMA) is increasingly recognized for its role in evaluating the integrity of the developing nervous system and predicting motor dysfunctions, particularly in conditions such as cerebral palsy (CP). However, the necessity for highly trained professionals has hindered the adoption of GMA as an early screening tool in some countries. In this study, we propose a deep learning-based motor assessment model (MAM) that combines infant videos and basic characteristics, with the aim of automating GMA at the fidgety movements (FMs) stage. MAM demonstrates strong performance, achieving an Area Under the Curve (AUC) of 0.967 during external validation. Importantly, it adheres closely to the principles of GMA and exhibits robust interpretability, as it can accurately identify FMs within videos, showing substantial agreement with expert assessments. Leveraging the predicted FMs frequency, a quantitative GMA method is introduced, which achieves an AUC of 0.956 and enhances the diagnostic accuracy of GMA beginners by 11.0%. The development of MAM holds the potential to significantly streamline early CP screening and revolutionize the field of video-based quantitative medical diagnostics.

The Prechtl General Movements Assessment (GMA) serves as a valuable tool for assessing the developmental status of an infant's nervous system and the potential presence of motor abnormalities[1]. It accomplishes this by evaluating the quality of general movements (GMs), which are part of the infant's repertoire of spontaneous movements[1,2]. Typically, when infants reach a corrected age of 9–20 weeks, GMs exhibit characteristics such as moderate speed, variable acceleration, and engagement of the neck, trunk, and limbs in various directions[1–3]. These specific movements are referred to as fidgety movements (FMs).

Notably, the absence of FMs represents a highly reliable indicator for predicting the early onset of cerebral palsy (CP)[4], which is the most common motor disability in childhood, encompassing a group of disorders that profoundly affect an individual's movement, balance, and posture[5–7]. CP affects approximately 1.4–2.5 cases per 1000 live births in high-income countries, with a higher incidence observed in low- and middle-income countries[8,9]. GMA plays a pivotal role in facilitating early screening for CP, enabling the detection of potential risks. This, in turn, allows for the optimization of the infant's malleable

[1]State Key Laboratory of Microbial Metabolism, Joint International Research Laboratory of Metabolic and Developmental Sciences, Department of Bioinformatics and Biostatistics, School of Life Sciences and Biotechnology, Shanghai Jiao Tong University, Shanghai, China. [2]SJTU-Yale Joint Center of Biostatistics and Data Science, National Center for Translational Medicine, MoE Key Lab of Artificial Intelligence, AI Institute, Shanghai Jiao Tong University, Shanghai, China. [3]Department of Health Management, Shanghai Children's Hospital, School of Medicine, Shanghai Jiao Tong University, Shanghai, China. [4]Shanghai Engineering Research Center for Big Data in Pediatric Precision Medicine, NHC Key Laboratory of Medical Embryogenesis and Developmental Molecular Biology & Shanghai Key Laboratory of Embryo and Reproduction Engineering, Shanghai Jiao Tong University, Shanghai, China. [5]School of Medicine, The Chinese University of Hong Kong, Shenzhen, Guangdong, China. [6]These authors contributed equally: Qiang Gao, Siqiong Yao. ✉e-mail: gjyu@shchildren.com.cn; huilu@sjtu.edu.cn

brain development and the minimization of the adverse effects of motor impairments through personalized interventions[7,10]. Health professionals trained in GMA possess the ability to qualitatively assess FMs by observing infants in a supine position, free from crying or external stimuli. They categorize an infant's GMs at the FMs stage as normal FMs, abnormal FMs, or absent FMs[1,2,11], in which the absent FMs demonstrate a sensitivity of 98% and a specificity of 94% in CP prediction[12,13]. Nevertheless, GMA still requires highly trained and certified professionals, as well as experience and regular calibration, to achieve the desired precision and consistency[4,14]. Consequently, the widespread use of GMA in the general population is not feasible, necessitating the exploration of alternative automated GMA methods.

Artificial intelligence has significantly advanced the assessment of neurodevelopmental deficits in infants through motor assessment[15,16]. The use of wearable sensor devices and motion capture markers has enabled the collection of precise motion data, which can then be harnessed to predict various types of GMs, CP, or other motor impairments using machine learning techniques[17–22]. However, it is worth noting that these devices can interfere with the normal movements of infants and impose demanding operational requirements on medical staff[15]. In contrast, video-based computer vision methods, such as background subtraction or optical flow algorithms, offer a non-invasive means of detecting changes between consecutive frames. This approach enables the representation of overall motion changes or the tracking of specific limb motion changes without disturbing the infant's movements[15,16,23–27]. The advent of deep learning has revolutionized computer vision techniques, particularly through pose estimation methods that allow for the direct extraction of an infant's joint coordinates from video data. This innovation helps overcome challenges related to illumination, background interference, and other sources of noise[28]. These estimated coordinates serve as valuable inputs for extracting motion representation information, which can then be utilized in traditional machine learning or deep learning-based spatiotemporal models. This integration has led to a significant improvement in the accuracy of predicting specific neurological deficits[29–33].

In the context of automated GMA methods at the FMs stage, high accuracy is typically achieved through qualitative approaches that leverage spatiotemporal models. However, these methods often provide only the final classification outcome for GMs[31,32]. In contrast, quantitative tools offer objective measurements and numerical data, reducing the influence of subjective interpretation and enabling a more comprehensive assessment of a patient's condition[34,35]. Some studies have attempted to quantify GMA by analyzing the motion patterns in various body parts, including the head, trunk, arms, and legs[29,30]. These analyses consider factors such as direction, magnitude, speed, and acceleration, along with their statistical characteristics, to derive quantitative values[25,26,29,30]. Nevertheless, quantitative approaches have not consistently matched the performance of qualitative methods in GMs classification, and the emphasis on different body parts or various indicators contradicts the spatial integrity in gestalt perception[36]. Moreover, both existing qualitative and quantitative methods often lack the interpretability required for their users to fully comprehend the model results. While some studies have acknowledged the occurrence of FMs, their efforts to explain the precise locations where FMs appear have been insufficient[31,32].

In this study, we employ a 3D pose estimation method to predict the coordinates of critical joints in infant videos and introduce a distance representation approach to capture the overall motion patterns. Building upon the principles of multi-instance learning (MIL)[37] and Transformers-based techniques[38], we present an infant motor assessment model (MAM). MAM utilizes infant movements and basic characteristics to achieve precise predictions of GMs at the FMs stage. To enhance the model's compliance with GMA principles while maintaining robust interpretability, we designed a dedicated FMs reference branch and a specialized Closeness loss function. When combined with MIL framework, these components enable our model to identify the presence of FMs within the videos. Furthermore, we propose a quantitative GMA diagnosis method at the FMs stage based on the predicted frequency of FMs. This approach has been validated as effective and demonstrated to improve the diagnostic accuracy for GMA beginners. Given the substantial predictive value of the absence of FMs in detecting CP, our automated approach represents a significant contribution to advancing the universal early screening of CP.

## Results

### GMA results and video information

This study comprises three cohorts, and the cohort filtering process is visually represented in Fig. 1. Cohort 1 and Cohort 2 are primarily utilized for internal cross-validation and external validation, respectively, while Cohort 3 functions as a reference and is employed for pre-training the MAM. Each infant included in this study is accompanied by a video that is pivotal for assessing their GMs. Infants whose corrected ages fall outside the range of 9–20 weeks, those lacking essential basic characteristics, or those whose corresponding videos do not adhere to the recording requirements stipulated by the GMA have been systematically excluded from Cohort 1 and Cohort 2. Additionally, infants displaying abnormal FMs have been excluded, as this category is rare and possesses limited predictive power[2,11]. It is worth noting that normal FMs can be further subdivided into continuous FMs, intermittent FMs, and sporadic FMs. The prognosis associated with sporadic FMs closely resembles that of absent FMs[39,40]. Consequently, we have categorized infants with continuous FMs and intermittent FMs as the normal group, and infants with sporadic FMs and absent FMs as the risk group. Following these exclusions and categorizations, the internal cross-validation dataset comprises 691 infants in the normal group and 215 infants in the risk group. The external validation dataset consists of 173 infants in the normal group and 48 infants in the risk group. Comprehensive details regarding the basic characteristics of the infants included in both the internal and external datasets are provided in Table 1. To assess potential differences between the normal and risk groups, we conducted a Chi-square test to examine sex distribution, while other essential characteristics, such as gestational age, birth weight, and corrected age, were analyzed using either the $t$-test or Mann–Whitney test, depending on the normality of the data. Notably, no statistically significant differences ($p > 0.01$ for all comparisons) were observed between these two groups. Cohort 3 serves as a repository for infants excluded by Cohort 1 solely due to the absence of basic characteristics. The videos of these infants are meticulously segmented into FMs or non-FMs clips. These clips are employed as references and used for pre-training MAM, contributing to the model's robustness and accuracy.

Within Cohort 1, the median video duration is 295 s, with a range of 119 to 653 s. Frame rates predominantly include 25 fps (95.4%), with a smaller proportion at 29 fps (4.6%). Pixel resolutions encompass 720 × 576 (1.7%), 1280 × 720 (4.9%), 1440 × 1080 (1.3%), and 1920 × 1080 (92.2%). In Cohort 2, the median video duration is 297 s, with a range of 181 to 556 s. Frame rates are all 25 fps (100%), and pixel resolutions encompass 1280 × 720 (10.4%), 1440 × 1080 (5.4%), and 1920 × 1080 (89.6%). Cohort 3 exhibits a median video duration of 299 s, ranging from 188 to 409 s. Frame rates in this cohort are predominantly 25 fps (96.3%), with a smaller proportion at 29 fps (3.7%). Pixel resolutions encompass 720 × 576 (2.9%), 1280 × 720 (16.9%), 1440 × 1080 (2.1%), and 1920 × 1080 (78.2%).

### Performance of MAM in GMs prediction

As shown in Fig. 2, MAM is organized into three key components: the Ref Branch, the Main Branch, and the Info Branch. The performance of MAM in GMs prediction at the FMs stage was tested both in internal and external datasets. The receiver operating characteristic (ROC)

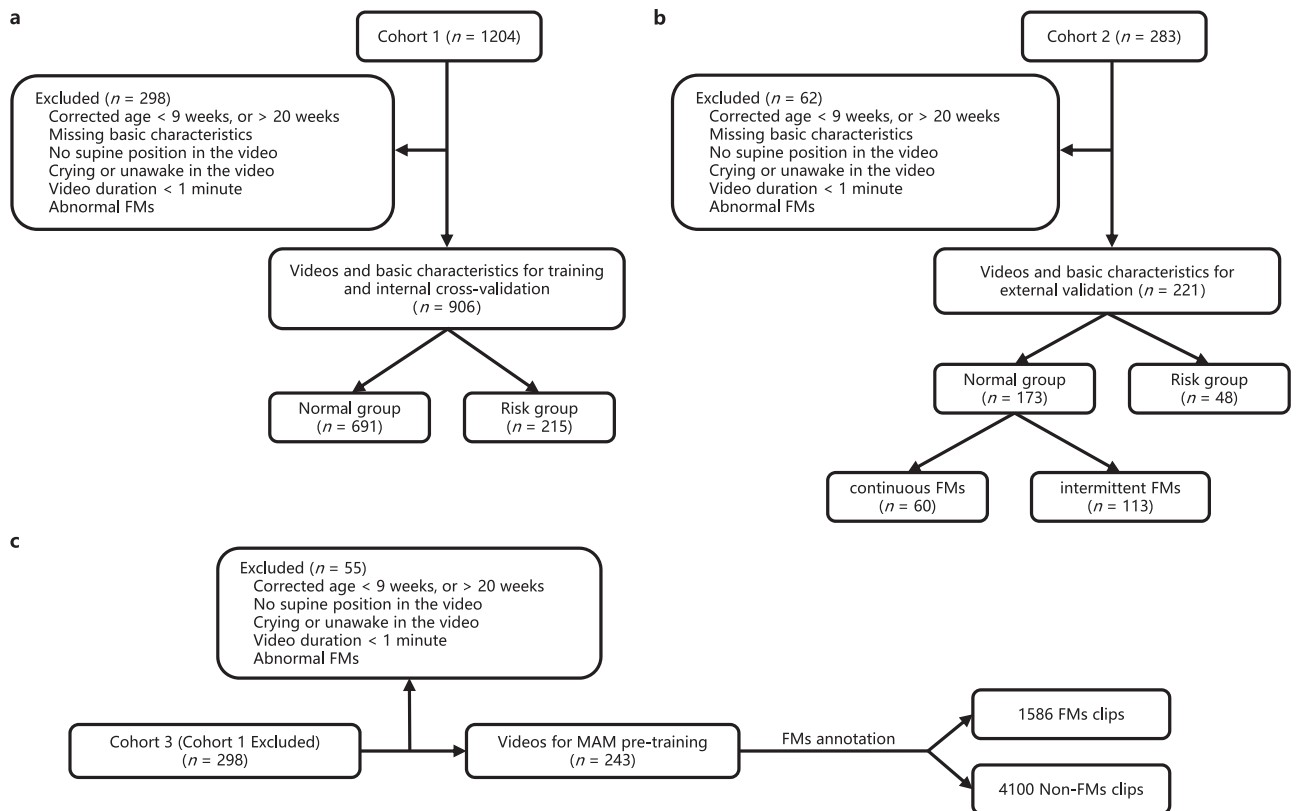

**Fig. 1 | The flowchart of cohort filtering. a** Cohort 1 initially comprised 1204 infants. 298 infants were excluded, and 906 infants were retained for training and internal cross-validation, including 691 infants categorized as having normal FMs and 215 infants categorized as having risk FMs (absent FMs or sporadic FMs). **b** Cohort 2 initially comprised 283 infants. 62 infants were excluded, and 221 infants were retained for external validation, including 173 infants categorized as having normal FMs and 48 infants categorized as having risk FMs. The infants in normal group included 60 infants having continuous FMs and 113 infants having intermittent FMs. **c** Cohort 3 is the 298 infants excluded from Cohort 1. 55 infants were excluded due to age, video quality or abnormal FMs, and 243 infants were retained for MAM pre-training, whose videos were annotated with FMs and further segmented into 1,586 FMs clips and 4100 non-FMs clips. FMs fidgety movements.

**Table 1 | The basic characteristics of the infants in internal and external datasets**

| | Internal cross-validation dataset (n = 906) | | | External validation dataset (n = 221) | | |
|---|---|---|---|---|---|---|
| | Normal group | Risk group | *p* value | Normal group | Risk group | *p* value |
| *n* | 691 (76.3%) | 215 (23.7%) | | 173 (78.3%) | 48 (21.7%) | |
| Sex | | | | | | |
| Male | 353 (51.1%) | 114 (53.0%) | 0.676 | 87 (50.0%) | 27 (56.3%) | 0.570 |
| Female | 338 (48.9%) | 101 (47.0%) | | 86 (50.0%) | 21 (43.8%) | |
| GA (week) | 35.11 (3.22) | 35.68 (3.22) | 0.012 | 35.14 (3.34) | 35.58 (3.38) | 0.384 |
| BW (g) | 2415.67 (665.88) | 2514.93 (699.11) | 0.093 | 2414.86 (687.62) | 2526.56 (671.03) | 0.313 |
| CA (week) | 12.53 (2.72) | 12.53 (2.64) | 0.866 | 12.73 (2.72) | 12.75 (2.86) | 0.928 |

Data are represented by *n* (%) or mean (sd). The *p* values are calculated using Chi-square test, *t*-test or Mann–Whitney test. All tests are two-tailed.
*GA* gestational age, *BW* birth weight, *CA* corrected age.

curve of each fold in internal cross-validation and mean ROC curve in external validation are shown in Supplementary Figs. 3 and 4. MAM achieved an area under ROC curve (AUC) value of 0.973 (±0.007), an accuracy of 0.938 (±0.007), a sensitivity of 0.939 (±0.021), a specificity of 0.934 (±0.014), a positive predictive value (PPV) of 0.826 (±0.031) and a negative predictive value (NPV) of 0.980 (±0.006) in the internal dataset. In the external validation, MAM achieved an AUC value of 0.967 (±0.005), an accuracy of 0.934 (±0.008), a sensitivity of 0.925 (±0.024), a specificity of 0.936 (±0.009), a PPV of 0.802 (±0.022) and a NPV of 0.978 (±0.008).

We then compared the performance of MAM with other state-of-the-art methods, as shown in Table 2. Using the same video data and

parameter adjustment way, MAM far exceeds other methods in terms of AUC, accuracy, sensitivity, specificity and PPV, regardless of whether in internal validation or external validation. All models have relatively high NPV, but MAM still achieves the best performance.

Furthermore, we observed the performance of the MAM without the Info Branch (MAM.w/o.info). Compared with the original MAM, only the average specificity and average PPV of MAM.w/o.info in internal validation show slightly better, while the averages of other metrics are lower in both internal and external validation. However, after conducting the Mann-Whitney test, it was found that there was no significant difference (*p* > 0.01 for all comparisons) in all metrics between MAM and MAM.w/o.info. Furthermore, SHapley Additive

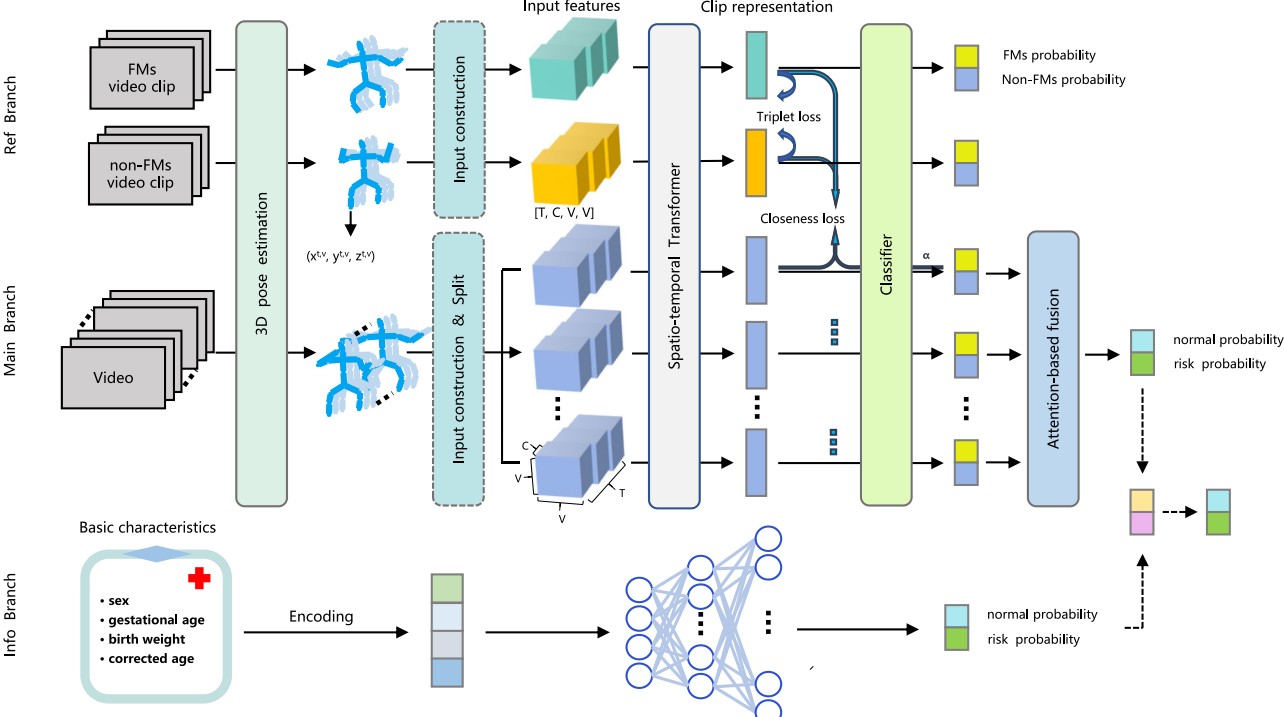

**Fig. 2 | The overall architecture of MAM.** MAM consists of a Ref Branch, a Main Branch, and an Info Branch. The Ref Branch, containing a 3D pose estimation step, an input construction step, a spatio-temporal Transformer (Supplementary Fig. 1) and a classifier, employs small FMs or non-FMs clips as inputs, and outputs FMs and non-FMs probability. The Main Branch, with an additional split step and an attention-based fusion (Supplementary Fig. 2), employs whole videos as inputs, and outputs normal and risk probability. The Info Branch employs basic characteristics as input, and output normal and risk probability by a fully connected network. The normal and risk probability given by the Main Branch and Info Branch are combined to give the final normal and risk probability.

## Table 2 | The performance of MAM in internal cross-validation and external validation

| Method | AUC | Accuracy | Sensitivity | Specificity | PPV | NPV |
|---|---|---|---|---|---|---|
| **Internal cross-validation** | | | | | | |
| EML | 0.846 ± 0.034 | 0.787 ± 0.017 | 0.786 ± 0.078 | 0.788 ± 0.039 | 0.556 ± 0.033 | 0.918 ± 0.026 |
| STAM | 0.880 ± 0.005 | 0.840 ± 0.020 | 0.832 ± 0.030 | 0.842 ± 0.036 | 0.643 ± 0.048 | 0.938 ± 0.008 |
| WO-GMA | 0.912 ± 0.010 | 0.856 ± 0.017 | 0.879 ± 0.042 | 0.848 ± 0.025 | 0.663 ± 0.034 | 0.955 ± 0.015 |
| MAM | **0.973 ± 0.007** | **0.938 ± 0.007** | **0.939 ± 0.021** | 0.934 ± 0.014 | 0.826 ± 0.031 | **0.980 ± 0.006** |
| MAM.w/o.info | 0.965 ± 0.006 | 0.931 ± 0.012 | 0.912 ± 0.010 | **0.937 ± 0.016** | **0.832 ± 0.034** | 0.969 ± 0.003 |
| **External validation** | | | | | | |
| EML | 0.844 ± 0.026 | 0.767 ± 0.027 | 0.850 ± 0.035 | 0.744 ± 0.039 | 0.483 ± 0.037 | 0.947 ± 0.010 |
| STAM | 0.882 ± 0.011 | 0.810 ± 0.016 | 0.879 ± 0.018 | 0.791 ± 0.025 | 0.539 ± 0.025 | 0.959 ± 0.004 |
| WO-GMA | 0.906 ± 0.014 | 0.848 ± 0.019 | 0.904 ± 0.048 | 0.832 ± 0.035 | 0.603 ± 0.040 | 0.970 ± 0.013 |
| MAM | **0.967 ± 0.005** | **0.934 ± 0.008** | **0.925 ± 0.024** | **0.936 ± 0.009** | **0.802 ± 0.022** | **0.978 ± 0.008** |
| MAM.w/o.info | 0.966 ± 0.006 | 0.928 ± 0.008 | 0.908 ± 0.038 | 0.933 ± 0.005 | 0.790 ± 0.013 | 0.974 ± 0.011 |

Data are represented by mean ± sd. The highest value for each metric is shown in bold.
*EML* ensemble machine learning model by Mccay et al.[30], *STAM* spatio-temporal attention-based model by Nguyen-Thai et al.[31], *WO-GMA* weakly supervised online action detection model by Luo et al.[32], *MAM* motor assessment model, *MAM.w/o.info* MAM without the Info Branch, *AUC* area under receiver operating characteristic curve, *PPV* positive predictive value, *NPV* negative predictive value.

exPlanations (SHAP)[41] method was applied to explore the contribution of the infants' characteristics in the Info Branch to the final prediction. As shown in Supplementary Fig. 5, we found that the video feature in the Main Branch dominated the prediction of normal probability, wherein the larger the video feature value after fusion, the greater the prediction probability of the normal group. The contribution degree of the four basic characteristics was small, and it was difficult to distinguish the direction of their influence on the prediction of normal probability.

To illustrate the role of distance matrices and 3D pose estimation (see Methods), we evaluated the AUC values obtained by the model using different input construction ways and pose estimation methods of different dimensions (Supplementary Table 1). We found that the proposed input construction way of distance matrices outperformed that of using coordinates, velocities and accelerations, or their combinations. In addition, under the same input construction way, the performance obtained by using 3D pose estimation method is superior to that obtained by using 2D pose estimation method.

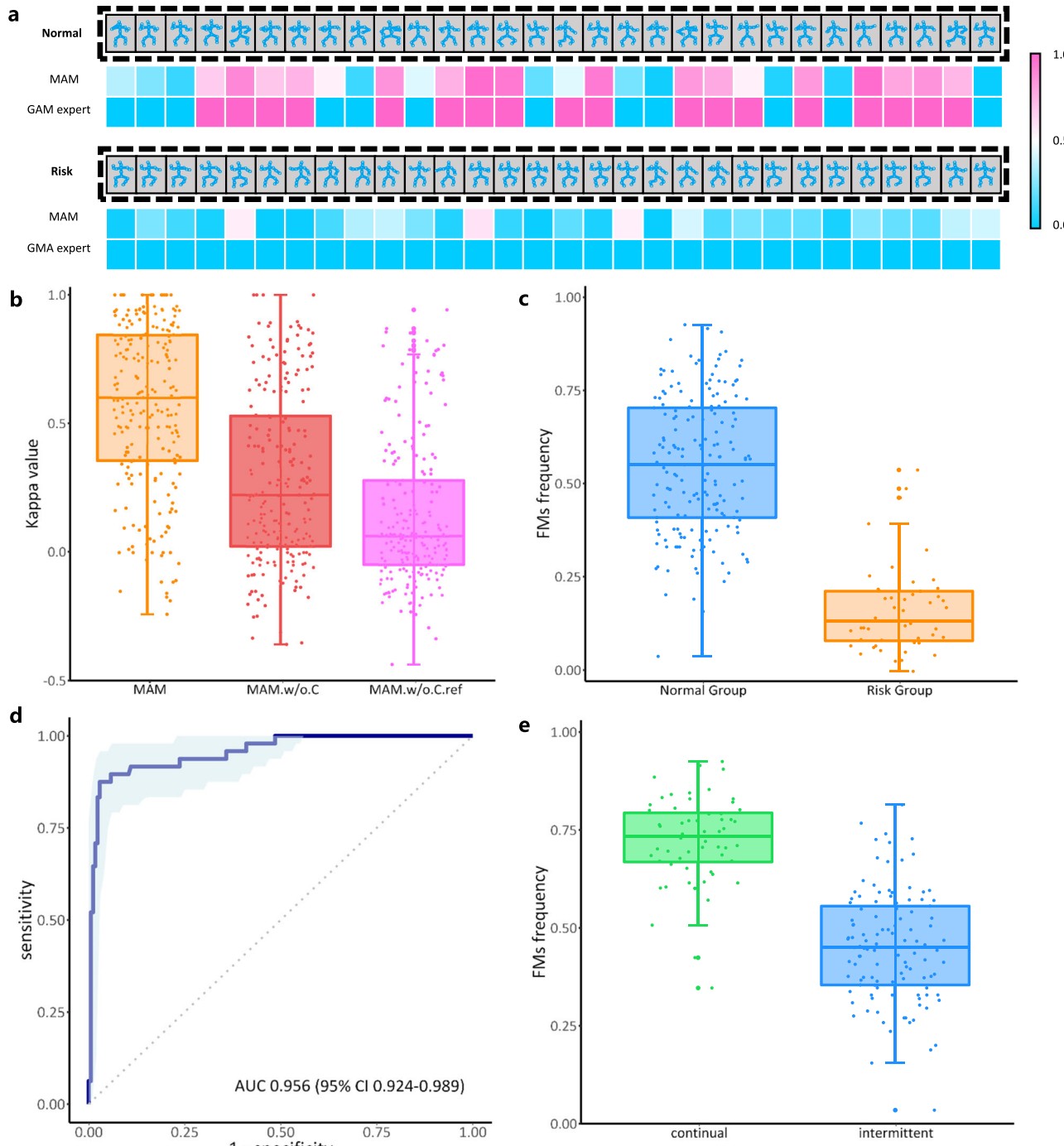

**Fig. 3 | The performance of MAM in FMs identification and quantification.**
**a** Examples of FMs and non-FMs clips evaluated by MAM and GMA experts. Each rectangle inside the black dashed box represents a 9.6-s video clip of the identified infant pose information. The bottom squares represent the corresponding predictions, with blue denoting a non-FMs clip and pink denoting an FMs clip. **b** Distribution of concordance in the evaluations of FMs in videos from the external validation dataset ($n = 221$) by GMA experts and different MAM types. MAM.w/o.C MAM without Closeness loss function, MAM.w/o.C.ref MAM without Closeness loss function and Ref Branch. **c** Distribution of predicted FMs frequencies by MAM in normal group ($n = 173$) and risk group ($n = 48$). **d** The receiver operating characteristic (ROC) curve for classification of normal and risk group using predicted FMs frequency by MAM. The light blue area represents the 95% confidence interval, with the center line representing the mean. AUC area under ROC curve, **e** Distribution of predicted FMs frequencies by MAM in infants having continual FMs ($n = 60$) and intermittent FMs ($n = 113$). For all boxplots in **b**, **c**, **e**, the center lines of boxplots indicate the median values; box limits show upper and lower quartiles; whiskers extend from box limits to the farthest data point within 1.5 × interquartile range; points beyond whiskers are outliers. Source data are provided as a Source Data file.

## Performance of MAM in FMs identification and quantification

To investigate the capability of MAM in identifying FMs, we converted the FMs probabilities obtained by the classifier in the Main Branch into FMs clips or non-FMs clips setting 0.5 as the threshold. We then compared evaluations for FMs or non-FMs clips in the entire video by the MAM and the GMA experts. Figure 3a shows examples of a normal infant's video and a risk infant's video, with each clip being assigned a value from 0 to 1 by the MAM. The closer a clip's value is to 1, the more

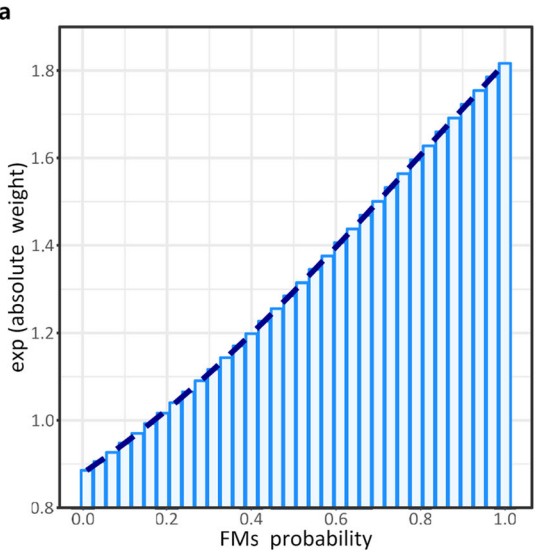

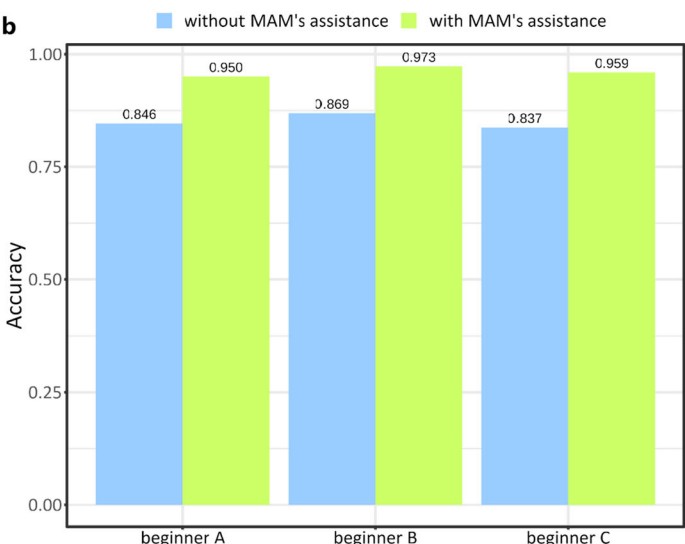

**Fig. 4 | The prediction details of MAM and MAM's assistance to GMA beginners.** **a** Absolute contribution of FMs probability to the normal probability. The exponential function in the softmax operation is taken to better reflect the degree of contribution. **b** Comparison of diagnostic accuracy among three GMA beginners with or without MAM's assistance. Source data are provided as a Source Data file.

likely it is to be an FMs clip. The experts' results are represented as either 0 or 1, with 0 denoting a non-FMs clip and 1 denoting an FMs clip. In the examples, the concordance of the judgments between MAM and GMA experts on FMs is commendable. To provide a more comprehensive view, we drew a boxplot to reflect the distribution of concordance between MAM and GMA experts across all external validation data. As shown in Fig. 3b (left), the median kappa value achieved 0.601 (Q1–Q3 0.357–0.845). The Closeness loss function (see Methods) and the Ref Branch were effective in enhancing MAM's ability to identify FMs. Without using the Closeness loss function, the concordance, shown in Fig. 3b (middle), reached a kappa value of 0.224 (Q1–Q3 0.025–0.530). Without using the Closeness loss function and Ref Branch, the concordance dropped to a kappa value of 0.064 (Q1–Q3 − 0.046–0.280), as shown in Fig. 3b (right).

We calculated the proportion of FMs clips in an entire video and named it "FMs frequency". As shown in Fig. 3c, the median of FMs frequency in the external validation dataset is 0.553 (Q1–Q3 0.412–0.706) in the normal group and 0.135 (Q1–Q3 0.082–0.215) in the risk group. There is a significant difference ($p < 0.01$) in FMs frequency between the normal group and the risk group. Further, we classified videos of the normal group and risk group directly using the FMs frequency to explore if MAM could transform GMA from a qualitative to a quantitative tool. As shown in Fig. 3d, an AUC of 0.956 (95% CI 0.924–0.989) is obtained, indicating that we can quantify GMA in this way.

We then observed the distribution of FMs frequency evaluated by MAM in infants having continuous FMs and intermittent FMs. As shown in Fig. 3e, the median FMs frequency in infants having continuous FMs is 0.738 (Q1–Q3 0.672–0.798), and the median FMs frequency in infants having intermittent FMs is 0.455 (Q1–Q3 0.359–0.560). There is a significant difference ($p < 0.01$) in FMs frequency between these two categories. Using a threshold of 0.603 to classify continuous FMs and intermittent FMs based on FMs frequency, results in an AUC of 0.932 (95% CI 0.892–0.973) and an accuracy of 0.902. These results further validate that our approach can quantitatively automate GMA.

### MAM's ability to assist GMA beginners

The attention-based fusion (Supplementary Fig. 2) step in the Main Branch learns the contribution degree of the FMs probability of each "instance" to predict the normal probability of the "bag" (see Methods). For the trained MAM, the absolute contribution of each "instance" to the normal probability of the "bag" solely depends on its FMs probability, whereas the relative contribution also depends on other "instances" in the "bag" due to the softmax operation. Figure 4a displays the absolute weights (exponentially processed) of the "instances" for the final normal probability prediction at different FMs probabilities. The curve indicates that clips with higher FMs probability have greater value in predicting the normal probability of the whole video.

In clinical practice, the health professionals need to distinguish FMs from various other movements that occur concurrently with FMs, such as wiggling-oscillating and saccadic arm movements[2,11]. However, this is difficult for some less-experienced observers[2,11]. We hypothesized that the clips featuring high FMs probability in the videos predicted by MAM may provide valuable reference points for beginners, helping them build confidence in assessing developing infants. Therefore, we examined MAM's capacity to assist beginners, primarily achieved through repeated display of focused clips with high FMs probability determined by the MAM.

Using GMA expert evaluations as reference, we compared the diagnostic accuracy of three GMA beginners (GMA certified for less than six months) with and without the assistance of MAM on the external validation dataset. As shown in Fig. 4b, without the assistance of MAM, the prediction accuracy of the three beginners was 0.846, 0.869, 0.837, respectively. However, with the assistance of MAM, their accuracy improved to 0.950, 0.973, and 0.959, respectively, with an average increase of 11.0%.

### Discussion

In this study, we developed a multi-instance multimodal motor assessment model based on the Transformer architecture to realize prediction of GMs in infants at the FMs stage, which aimed at expediting the early detection of CP. The proposed model referred to as MAM, outperforms extant methodologies across all evaluated metrics. Unlike existing methods[29–31] which exclusively relied on motion posture from videos to predict GMs, we adopted a multimodal fusion approach to explore the influence of infants' basic characteristics on the final prediction accuracy. Through hypothesis testing and SHAP analysis, it was discerned that these basic characteristics exerted a

marginal impact on the model's final predictive ability, where video features remained as the primary contributors. This also explains why health professionals typically do not consider additional information about the infant when assessing infants of a specific age group[1,11].

The perception of FMs is the prerequisite for health professionals to make correct GMs classification judgment at the FMs stage, and the absence of FMs constitutes the most robust indicator for predicting CP[4,40]. Therefore, if the model can pinpoint the location of FMs within videos, it not only provides decision support for health professionals but also enhances the model's transparency and reliability. While previous methods[31,32] have taken note of this aspect, they fell short of achieving notable advancement. Leveraging MIL, referencing the Ref Branch, and integrating the specially devised Closeness loss function, MAM effectively identifies the location of FMs. Figure 3b highlights the crucial role of the Ref Branch and Closeness loss function. The original and complete MAM achieves substantial agreement with GMA experts in identifying FMs. Additionally, the clips where MAM assigns high FMs probability have been shown to be valuable to GMA beginners, as they contribute to an improvement in diagnostic accuracy.

Quantification has always been a focus in medical diagnostic tools, as it provides more objective, intuitive, and convenient evidence for both health professionals and patients[34,35]. However, the classification reliability of quantitative GMA at the FMs stage is not adequate[25,26]. Moreover, previous quantitative methods have deviated from the gestalt principle of GMA[29,30]. Here, based on the FMs identified by MAM, we proposed quantifying GMA using the concept of "FMs frequency", which was expressed as the proportion of FMs clips. Without breaking down the individual limb movements of the infant, this approach adheres to the gestalt principle of maintaining spatial integrity during observation. We further verified that this approach was able to distinguish between normal and risk group videos, and between continual and intermittent FMs videos. These encouraging results demonstrate the effectiveness of our quantitative GMA method.

Early screening of CP ensures that the interventions can be implemented at critical stages of brain development to maximize motor and cognitive outcomes, and reduce the incidence of other co-morbidities such as visual and hearing impairments[7,13]. Additionally, early screening provides parents with a clear result for their high-risk baby, reducing their stress and anxiety, and enhancing their coping abilities[7,42]. Among the risk group in Cohort 1, 156 infants retained their follow-up results at 2–4 years of age, of which 129 (82.7%) were eventually diagnosed with CP. The majority of these infants have received timely intervention after GMA, so the proportion of infants who were diagnosed as only having mild CP was 88.4%. This highlights the significance of GMA in early screening and enabling early intervention. Yet, GMA is still plagued by limitations such as requirement for training and certification, as well as being labor-intensive[4]. Our MAM method aptly addresses these issues. MAM can also serve as an aid for GMA beginners, helping them boost their prediction accuracy. For low- and middle- income countries and regions that lack GMA training, MAM can act as an alternative solution to GMA. Additionally, MAM's high sensitivity allows it to serve as a screening tool, making it possible to use GMA at a population level.

There are some limitations of the study. First, the infants whose videos were used in this study were high-risk infants. Thus, the effect of the model on normal infants cannot be directly assessed. Second, all the videos in this study were from hospitals. Therefore, the results do not directly reflect the MAM's performance in video recordings outside hospitals, such as those shot at home. In the future, MAM should be tested with non-high-risk infants and in diverse scenarios.

In conclusion, we have developed a quantitative and explainable motor assessment model based on deep learning to automate GMA and facilitate early screening of CP. The MAM demonstrated good performance while adhering to the principles of GMA. Our study provides a possible paradigm for video-based quantitative medical diagnostics.

## Methods
### Description of datasets
This study was approved by the Ethics Review Committee at Shanghai Children's Hospital, Shanghai Jiao Tong University. Parents of all infants in the study provided written informed consent prior to their infants' inclusion, and the videos and basic characteristics were anonymized. The Cohort 1 contains 1204 distinct high-risk infants (e.g., preterm birth, low birth weight, respiratory distress syndrome, in vitro fertilization, atrial septal defect, and hyperbilirubinemia) from Shanghai Children's Hospital collected from January 2013 to June 2019. Cohort 2 contains 283 distinct high-risk infants from Shanghai Children's Hospital collected from 2019 July to February 2023. The flow-chart of cohort filtering is shown in Fig. 1. 906 infants were retained in Cohort 1 for internal cross-validation, and 221 infants were retained in Cohort 2 for external validation. Cohort 3 is the 298 infants excluded from Cohort 1, in which 55 infants were excluded due to age, video quality or abnormal FMs, and 243 infants were retained for pre-training MAM.

A video recording of each infant according to GMA standards was obtained. The videos were annotated by two GMA experts blinded to the infants' medical histories. Both experts held GMA certification and had over 5 years of assessment experience. The experts made independent judgments, but were allowed to discuss and reach consensus in case of inconsistent results. The reliability between the two experts reached a kappa value of 0.947. In Cohort 1, 691 infants were categorized into the normal group and 215 infants were categorized into the risk group. In Cohort 2, 173 infants were categorized into the normal group (60 continuous FMs and 113 intermittent FMs) and 48 infants were categorized into the risk group. In Cohort 3, all videos were annotated for the intervals of FMs appearance. We cropped 1586 FMs clips that are completely covered by FMs and 4100 non-FMs clips that contain no FMs, each 9.6 s long. All the videos in Cohort 2 were also annotated for the intervals of FMs appearance. These videos were further segmented into 9.6-second clips with a step size of 6 s. Based on whether the proportion of FMs within each clip was greater than 0.5, these clips were categorized as FMs clips or non-FMs clips. This categorization was used to assess the concordance between MAM and GMA experts in identifying FMs.

### Overall architecture of MAM
MAM represents a multi-instance multimodal model founded on the Transformer architecture. As depicted in Fig. 2, MAM is organized into three key components: the Ref Branch, the Main Branch, and the Info Branch. The Ref Branch is specifically designed for pre-training the spatio-temporal Transformer (Supplementary Fig. 1) and the classifier within MAM. It utilizes video clips featuring FMs and non-FMs from Cohort 3. The training process incorporates the Triplet loss[43], a form of margin loss, to distinguish between FMs clip representations and non-FMs clip representations. These representations are dynamically adapted during training and serve as crucial references for the clip representations in the Main Branch. Both the Main Branch and Ref Branch share a substantial portion of their architectural elements. The fine-tuned 3D pose estimation model is responsible for extracting critical joint coordinates from supine infant videos. Importantly, the input construction step involves distance matrices of each dimension, which encapsulate overall motion patterns and capture potential coordination relationships among non-adjacent joints. The Main Branch incorporates an additional split step, dividing the input (referred to as a "bag" in MIL) into small parts ("instance" in MIL) of the same size as those in the Ref Branch. Meanwhile, an extra attention-based fusion (Supplementary Fig. 2) step is responsible for amalgamating the probabilities of FMs and non-FMs instances into the normal

and risk probability of the "bag". The designed Closeness loss function enables instances' clip representations to dynamically converge towards the FMs or non-FMs representation centers of the Ref Branch, resulting in improved instance prediction outcomes. The Info Branch utilizes the basic characteristics (sex, gestational age, birth weight, and assessment age) of the infants corresponding to the videos in the Main Branch for predictions. The resultant predictions are integrated with those from the Main Branch to yield multimodal predictions for the final normal and risk probability.

### 3D information acquisition and pre-processing

We used the VideoPose3D[44] model to perform monocular 3D pose estimation of the infants. This model was trained on adult datasets[45]. However, the proportions of infant limbs are very different from those of adults, and the infants are in the supine position rather than the standing position of adults. Directly applying the original model to the infant videos will lead to a large bias in pose estimation. Therefore, we extracted 5 frames from each of the 298 videos in Cohort 3 and manually annotated the critical joints to fine-tune HRNet[46], which can be considered a kernel of VideoPose3D. Next, we migrated the kernel to the VideoPose3D framework and obtained 3D coordinates of the 17 critical joints in each frame of all videos in our dataset. The frame rate of the videos has been standardized to 25 fps before fed into the VideoPose3D model. Moving Average process was applied to the obtained coordinate values with a window size of 5 frames. We then calculated the mean and standard deviation of all coordinates in each of the three dimensions and used this information to normalize the coordinates.

### Input features construction

We constructed distance matrices between the joints in each frame in 3D to capture the potential coordination relationship between non-adjacent joints and represent the overall motion patterns, as shown below:

$$
\begin{bmatrix}
d_{1,1}^c & \cdots & d_{1,V}^c \\
\vdots & \ddots & \vdots \\
d_{V,1}^c & \cdots & d_{V,V}^c
\end{bmatrix}
$$

where $c$ is the coordinate dimension, $c \in \{1,2,\ldots,C\}$. $C$ is the number of coordinate dimensions, $C = 3$. $V$ is the number of critical joints of the infant in each frame, $V = 17$. $d_{i,j}^c$ is the Euclidean distance between the $i$-th critical joint and $j$-th critical joint in c dimension. The input features of each frame can be represented by a [$C$, $V$, $V$] tensor, while the input features of the whole video can be represented by a [$T^*$, $C$, $V$, $V$] tensor, where $T^*$ is the number of the frames.

### Definition of "bag" and "instance"

In MIL, the dataset consists of a set of "bags" with labels, wherein each "bag" contains several "instances" without labels[37]. With the aim of dividing the whole video in the Main Branch into small clips with durations of 9.6 s, which was consistent with the length of FMs and non-FMs clips in the Ref Branch, the split step in the Main Branch splits the [$T^*$, $C$, $V$, $V$] tensor into several [$T$, $C$, $V$, $V$] tensors with a step of $T_s$, where $T = 240$ and $T_s = 90$. Therefore, the [$T^*$, $C$, $V$, $V$] tensor of each input video in the Main Branch is considered a "bag", and the split [$T$, $C$, $V$, $V$] tensors are considered "instances".

### Construction details and loss functions of MAM

In the Ref Branch, the FMs and non-FMs clips are converted into [$T$, $C$, $V$, $V$] tensors after 3D pose estimation and input construction. Next, they go through the spatio-temporal Transformer to obtain the clip representations that integrate temporal and spatial information, and the classifier gives their FMs probabilities. Cross-entropy loss is calculated as follows:

$$
L_{c1} = -\frac{1}{n}\sum_{i=1}^{n}\left[g_i * \log(\hat{g}_i) + (1-g_i) * \log(1-\hat{g}_i)\right] \tag{1}
$$

where $g_i$ represents the true label with a value of 0 or 1, $\hat{g}_i$ represents the predicted FMs probability, and $n$ is the number of clips in one training batch in the Ref Branch. Before input to the classifier, the Triplet loss between the FMs clip representations and the non-FMs clip representations is calculated as follows:

$$
L_t^i = \begin{cases}
\max\left(\frac{1}{n_1}\sum_{k=1}^{n_1} D(\mathbf{a}_i, \mathbf{p}_k) - \frac{1}{n_2}\sum_{k=1}^{n_2} D(\mathbf{a}_i, \mathbf{n}_k) + \mathrm{margin}, 0\right), \mathbf{a}_i \in \{\mathbf{p}_k, k \in \{1,2,\ldots,n_1\}\} \\
\max\left(\frac{1}{n_2}\sum_{k=1}^{n_2} D(\mathbf{a}_i, \mathbf{n}_k) - \frac{1}{n_1}\sum_{k=1}^{n_1} D(\mathbf{a}_i, \mathbf{p}_k) + \mathrm{margin}, 0\right), \mathbf{a}_i \in \{\mathbf{n}_k, k \in \{1,2,\ldots,n_2\}\}
\end{cases} \tag{2}
$$

$$
L_t = \frac{1}{n}\sum_{i=1}^{n} L_t^i \tag{3}
$$

where $\mathbf{a}_i$ is an anchor that may be an FMs clip representation or a non-FMs clip representation, $\mathbf{p}_k$ is one of the $n_1$ FMs clip representations in the batch, and $\mathbf{n}_k$ is one of the $n_2$ non-FMs clip representations in the batch (where $n_1 + n_2 = n$), $D$ is the Euclidean distance function, and margin is a constant that controls the distance between the two classes. The role of this loss function is to shorten the distance between the same classes and increase the distance between different classes.

The Main Branch is the core of the MAM. An infant's input video successively goes through the 3D pose estimation and the input construction to generate the "bag" tensor. This tensor is then divided into "instance" tensors in [$T$, $C$, $V$, $V$] format by the split step. Next, the spatio-temporal Transformer converts these tensors into instance clip representations, and the classifier further gives their FMs prediction probabilities. We further constructed a Closeness loss function to dynamically adjust the position of the instance clip representations in the latent space, making them choose to be close to the FMs or non-FMs clip representation cluster. The formulas of the Closeness loss function are as follows:

$$
\overline{\mathbf{p}_k} = \frac{1}{n_1}\sum_{k=1}^{n_1} \mathbf{p}_k \tag{4}
$$

$$
\overline{\mathbf{n}_k} = \frac{1}{n_2}\sum_{k=1}^{n_2} \mathbf{n}_k \tag{5}
$$

$$
L_c^i = \begin{cases}
[\alpha_i * (1-\alpha_i)] * \dfrac{e^{\mathrm{Dist}(\mathbf{b}_i, \overline{\mathbf{p}_k})}}{e^{\mathrm{Dist}(\mathbf{b}_i, \overline{\mathbf{p}_k})} + e^{\mathrm{Dist}(\mathbf{b}_i, \overline{\mathbf{n}_k})}}, & \alpha_i \geq 0.5 \\
[\alpha_i * (1-\alpha_i)] * \dfrac{e^{\mathrm{Dist}(\mathbf{b}_i, \overline{\mathbf{n}_k})}}{e^{\mathrm{Dist}(\mathbf{b}_i, \overline{\mathbf{p}_k})} + e^{\mathrm{Dist}(\mathbf{b}_i, \overline{\mathbf{n}_k})}}, & \alpha_i < 0.5
\end{cases} \tag{6}
$$

$$
L_c = \frac{1}{n^*}\sum_{i=1}^{n^*} L_c^i \tag{7}
$$

where $\overline{\mathbf{p}_k}$ is the center of the FMs clip representation cluster, and $\overline{\mathbf{n}_k}$ is the center of the non-FMs clip representation cluster, $\mathbf{b}_i$ is an instance clip representation, $\alpha_i$ is the FMs prediction probability of $\mathbf{b}_i$, and $n^*$ is the number of instances in one training batch. Dist is the Euclidean distance function. As $\alpha_i$ changes with training, it causes the dynamic adjustment of the Closeness loss. Weighted summation of instance FMs probabilities is performed by the attention-based fusion module. The result is then processed by the softmax function to obtain the bag prediction of normal probability.

The Info Branch represents the processing of the basic characteristics of the infants. The encoded features are subjected to a fully connected neural network with nonlinear activation, followed by processing via the softmax function to obtain the predicted normal probability. Finally, the Main Branch and Info Branch predictions are fused to obtain the final normal probability. Cross-entropy loss is again calculated as follows:

$$L_{c2} = -\frac{1}{N}\sum_{i=1}^{N}\left[y_i * \log(\hat{y}_i) + (1 - y_i) * \log(1 - \hat{y}_i)\right] \quad (8)$$

where $y_i$ represents the true label with a value of 0 or 1, $\hat{y}_i$ represents the predicted normal probability, and $N$ is the number of bags (infant videos) in one training batch. The loss in the whole MAM is calculated as the sum of the Triplet loss, Closeness loss, and the two cross-entropy losses, which is represented by the following formula:

$$loss = L_t + L_c + L_{c1} + L_{c2} \quad (9)$$

The three branchers are trained together, and the weights of the spatio-temporal Transformer are shared in the Ref Branch and Main Branch. Thus, the whole MAM will be able to distinguish FMs and non-FMs clips on the premise that the judgment of the final normal probability is correct.

Supplementary Fig. 1 shows the structure of the spatio-temporal Transformer. The input tensors first pass through a 2D convolution kernel (Conv2D) of size [*V, 1*] to fuse the spatial connection information of each critical joint into itself. Next, the information between each critical joint is further exchanged via the spatial Transformer. The spatial Transformer utilizes 2 layers of Transformer encoder. The embedding dimension is 64, the number of attention heads is 4, and the number of feedforward network dimension is 256. Attention-based fusion is then implemented to integrate all spatial information, resulting in a sequence of feature vectors arranged in time. Position encoding is further added to this sequence to ensure that the temporal relationship of the feature vectors is not lost in the temporal Transformer. Similarly, information at each moment is exchanged through the temporal Transformer. The structure of the temporal Transformer is identical to that of the spatial Transformer. Finally, all temporal information is integrated by attention-based fusion to obtain the output vector.

Supplementary Fig. 2 shows the details of the attention-based fusion module. Each feature vector input to the module is processed by the linear layers and Tanh activation functions to obtain the weight value. All weight values are then processed by the softmax function to obtain the new weights, which add up to 1. Finally, the original input feature vectors are linearly combined according to the new weights to create fused output features.

### Hyperparameters
The instance clip length is 240 frames, and adjacent clips have 90 frames of overlap. The batchsize of the model training is 64, with half of it being FMs or non-FMs clip features and the other half being instance clip features in one normal group's bag and one risk group's bag. The model was trained 300 epochs with an initial learning rate of 0.001, using SGD as the optimizer and CyclicLR as the Scheduler. The Triplet loss margin is set as 0.4.

### Model application
In the validation and application phase, as shown in Supplementary Fig. 6, we used the Main Branch and Info Branch of the MAM to produce the normal and risk probability of each infant. The Ref Branch is abandoned and the attention-based fusion step in the Main Branch can be replaced by frequency calculations of FMs clips. Other parts are identical to the original MAM.

### Statistical analysis
All hypothesis tests were two-tailed and used a significance level of 0.01. For categorical variables, the Chi-Square test was employed, while for continuous variables, either a t-test or Mann–Whitney test was chosen based on the fulfillment of normality assumptions. Cohen's kappa statistic was used in measuring the reliability between the two GMA experts, and the concordance between MAM and GMA experts on FMs annotations.

The accuracy, sensitivity, specificity, PPV and NPV were used to measure the model performance. These metrics are defined as follows:

$$accuracy = \frac{N_{TP} + N_{TN}}{N_{TP} + N_{FP} + N_{TN} + N_{FN}} \quad (10)$$

$$sensitivity = \frac{N_{TP}}{N_{TP} + N_{FN}} \quad (11)$$

$$specificity = \frac{N_{TN}}{N_{TN} + N_{FP}} \quad (12)$$

$$PPV = \frac{N_{TP}}{N_{TP} + N_{FP}} \quad (13)$$

$$NPV = \frac{N_{TN}}{N_{TN} + N_{FN}} \quad (14)$$

where $N_{TP}$, $N_{TN}$, $N_{FP}$, $N_{FN}$ represent the number of true positive, true negative, false positive, false negative infants, respectively.

### Reporting summary
Further information on research design is available in the Nature Portfolio Reporting Summary linked to this article.

## Data availability
The datasets used in this study are available under restricted access due to privacy, ethical and legal considerations. Access can be obtained by contacting the corresponding author at gjyu@shchildren.com.cn who will provide a response within 14 days and supply the data use agreement limiting its use to non-commercial research purposes. Source data are provided with this paper.

## Code availability
The code of HRNet used in this paper is available in https://github.com/stefanopini/simple-HRNet.git. The code of VideoPose3D used in this paper is available in https://github.com/facebookresearch/VideoPose3D.git. The code of MAM is also available in https://github.com/qiang-Blazer/MAM. The statistical analyses and tests were down by R packages.

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

## Acknowledgements

This research is supported by National Key R&D Program of China (Grant No. 2022YFC2703103) to G.Y., Neil Shen's SJTU Medical Research Fund to H.L., SJTU Trans-med Awards Research STAR20210106 to H.L., the Innovative Research Team of High-Level Local Universities in Shanghai (SHSMU-ZDCX20212200) to H.L., and Open Project Program of National Research Center for Translational Medicine (Shanghai) (TMSK-2021-142) to G.Y.

## Author contributions

Q.G., S.Y., G.Y., and H.L. developed the concept for the manuscript. Q.G. and S.Y. contributed to drafting of the manuscript. Q.G. designed the model and analyzed the data. S.Y., G.Y., and H.L. contributed to critical revision of the manuscript. Y.T., C.Z., T.Z., and D.W. contributed to providing medical data and advice.

## Competing interests

The authors declare no competing interests.
