## [Peer Review File · Nature Communications]

Automating General Movements Assessment with quantitative deep learning to facilitate early screening of cerebral palsyREVIEWER COMMENTS

Reviewer #1 (Remarks to the Author):

The manuscript introduces a novel deep learning-based approach for automating the clinical observation technique of General Movement Assessment (GMA) using quantitative information from infants' spontaneous movements together with infants' basic characteristics (sex, birth weight, gestational age and assessment age). The proposed Motor Assessment Model (MAM) displays promising predictive accuracy in external validation. However, there are concerns that need to be addressed:

- In the abstract, the authors state that the proposed model can “diagnose CP in infants under five months of corrected age”. This is a bold statement considering that the model's diagnostic accuracy for CP has not been evaluated in the paper. The authors should revise the manuscript (including the title and abstract) to highlight the contribution of the manuscript towards automated GMA. CP is one of several disorders that could benefit from this progress, but the diagnostic accuracy for the specific disorder would then need to be assessed on cohorts with outcomes available.

- It is unclear from the manuscript how the consistency between predictions of MAM and GMA experts (presented in Figure 4B) was calculated. Please include details on what videos were used (e.g., the external validation dataset) and whether or not the GMA experts annotated FMs in all 10 second clips of these videos.

- Provide statistics on the video material, include range and median values for video length, spatial resolution and temporal resolution for each of the three cohorts.

- In “3D Information Acquisition”: describe the specific post-processing techniques used for smoothing and normalizing coordinate data along with their hyperparameters (e.g., window size of filter).

- In “Construction details and loss functions of MAM”: mention which distance function are used by the loss functions.

- Include more details on the Transformer models: how many layers in the Spatio Transformer and the Temporal Transformer, what are their embedding dimensions, how many attention heads etc.?

- I would recommend to present the main results for the predictive accuracy of MAM and state-of-the-art methods (Figure 3 in current version) in a table where positive and negative predictive values are included in addition to the existing values for AUC, accuracy, sensitivity and specificity.

- The current version of the manuscript requires polishing as it appears unorganized and unfinished:

- o Methodological descriptions in the Results section (e.g., L105-128: “Construction of a multi-instance multimodal motor assessment model, MAM”, L160-162: “We also applied the SHapley Additive exPlanations...”)

- o Subjective interpretations under Results (e.g., L141-143: “MAM achieved an extremely high area under the curve... Additionally, MAM's performance on the external validation dataset was also impressive...”)

- o Unreferenced claims (L70-71: “Although artificial intelligence-based automatic methods demonstrate high accuracy in predicting CP, they still fall short in achieving quantitative prediction.”)
- o Results that are not part of the Results section in the manuscript receive much attention in the beginning of the Discussion section (e.g., L233-239: “Further, we evaluated the AUC values obtained by the model using different input construction ways and pose estimation methods of different dimensions (Table S2)...”)
- o Incomplete figures (e.g., Figure 3: lacking legend for subpanel D and I would also suggest including labels in the subpanels indicating whether the results are from cross-validation or external validation, Figure S3: legend indicates that standard deviation is presented).

Reviewer #2 (Remarks to the Author):

This paper is reporting on a deep-learning based method for classification of General Movements, which is part of a screening process for CP. GMs assessment is the most reliable method to predict early cerebral palsy, but it is labour intensive as it is dependent on a trained individual scoring the video and it is not practical at a general population level. This is important as 50% of babies who go on to be diagnosed with CP have had an uneventful perinatal period and have gone home without admission to a Neonatal intensive Care unit or special care nursery.

There are several groups around the world investigating this and the authors report high levels of accuracy compared to previous studies. It’s important throughout the paper to highlight that this AI model is looking at GMA which is part of the CP screening process and results of this study will be further validated when the outcome of CP is known. It is possible that the children with absent fidgety movements have other neurodevelopmental impairments other than CP. Given the cohort was recruited between 2013-2019 it would be useful to include information about outcomes.

Summary

1. Cerebral Palsy is not usually referred to as a syndrome. Central dyskinesia is not usually in the definition.

Introduction

1. Justification around need for AI would be better supported by the need for scaling (page 3, line 49-50) rather than beginners. The method has good reliability when using the Prechtl method and doing training.

Results

1. Sporadic fidgety is usually classified as absent fidgety. There are 12 infants with sporadic fidgety in the FM+ category. This will affect the judgement and effectiveness of the tool.

2. It was difficult to be sure that the tool could differentiate between intermittent and sporadic fidgety movements.

Discussion

1. The results should be presented in relation to the GMA rather than CP, as you don't yet know how many of these children went on to have CP.

2. Through the manuscript the authors refer to "doctors" using GMA but it is used by a range of health professionals around the world and I would recommend "doctor" be replaced with "health professional"

3. On page 15, the authors write that Early diagnosis of CP is of the utmost importance. It would be important to explain the reasons for this which include supporting the suspicions of parents, being able to reassure them about outcomes and provide targeted early intervention to improve long term outcomes. This will also allow early screening for other co-morbidities such as visual and hearing impairments, enable active surveillance of hips and prevention of contractures and scoliosis.

Methods

1. Statistical analysis review is not included in this review. The journal may require separate advice on the statistical methodology and analysis and the mathematical equations from pages 15 to 19.

2. Further information is required on how the GMA scores were classified by humans. What methods of GMs was used – Prechtl or Hadders-Algra? There are references included to both methods. How did you define a "beginner" vs "GMA expert". What was the reliability between the two GMA experts?

Response to reviewers

Manuscript ID: NCOMMS-23-28586

Title: Deep learning-based quantitative motor assessment for early diagnosis of cerebral palsy

New title: Automating General Movements Assessment with quantitative deep learning to facilitate early screening of cerebral palsy

We sincerely thank the reviewers for their positive comments and constructive suggestions to improve our manuscript. Please note that as suggested by reviewer 1, we have changed the title of our manuscript to “Automating General Movements Assessment with quantitative deep learning to facilitate early screening of cerebral palsy”. As suggested by reviewer 2, we have grouped "sporadic FMs" and "absent FMs" into the same category. While these changes led to varied results, the overall outcome remains steady. The following are a copy of the comments (marked in blue) and our detailed point-by-point response. The manuscript content modified in response to the suggestions is indicated in orange in this file and is also highlighted in the revised manuscript.

Reply to Reviewer # 1

The manuscript introduces a novel deep learning-based approach for automating the clinical observation technique of General Movements Assessment (GMA) using quantitative information from infants' spontaneous movements together with infants' basic characteristics (sex, birth weight, gestational age and assessment age). The proposed Motor Assessment Model (MAM) displays promising predictive accuracy in external validation.

We greatly appreciate your positive and valuable comments.

However, there are concerns that need to be addressed:

- In the abstract, the authors state that the proposed model can “diagnose CP in infants under five months of corrected age”. This is a bold statement considering that the model’s diagnostic accuracy for CP has not been evaluated in the paper. The authors should revise the manuscript (including the title and abstract) to highlight the contribution of the manuscript towards automated GMA. CP is one of several disorders that could benefit from this progress, but the diagnostic accuracy for the specific disorder would then need to be assessed on cohorts with outcomes available.

We apologize for the bold statement, as our primary contribution is indeed automating and quantifying GMA at the FMs stage rather than directly predicting CP. We have changed the title from “Deep learning-based quantitative motor assessment for early diagnosis of cerebral palsy” to “Automating General Movements Assessment with quantitative deep learning to facilitate early screening of cerebral palsy”. The revised abstract is as follows: “The Prechtl General Movements Assessment (GMA) is increasingly recognized for its role in evaluating the integrity of the developing nervous system and predicting motor dysfunctions, particularly in conditions such as cerebral palsy (CP). However, the necessity for highly trained professionals has hindered the widespread adoption of GMA as an early screening tool. In this study, we propose a deep learning-based motor assessment model (MAM) that combines infant videos and basic characteristics, with the aim of automating GMA at the fidgety movements (FMs) stage. MAM demonstrates strong performance, achieving an Area Under the Curve (AUC) of 0.967 during external validation. Importantly, it adheres closely to the principles of GMA and exhibits robust interpretability, as it can accurately identify FMs within videos, showing substantial agreement with expert assessments. Leveraging the predicted FMs frequency, a quantitative GMA method is introduced, which achieves an AUC of 0.956 and enhances the diagnostic accuracy of GMA beginners by 11.0%. The development of MAM holds the potential to significantly streamline early CP screening and revolutionize the field of video-based quantitative medical diagnostics.” The introduction to GMA has also been moved to the beginning of the article: “The Prechtl General Movements Assessment (GMA) serves as a valuable tool for assessing the developmental status of an infant's nervous system and the potential presence of motor abnormalities¹. It accomplishes this by evaluating the quality of

general movements (GMs), which are part of the infant's repertoire of spontaneous movements^{1,2}. Typically, when infants reach a corrected age of 9-20 weeks, GMs exhibit characteristics such as moderate speed, variable acceleration, and engagement of the neck, trunk, and limbs in various directions^{1,2,3}. These specific movements are referred to as fidgety movements (FMs)."

Given that the sensitivity and specificity of GMA at FMs stage for predicting CP both exceed 90%^{1,2}, this research also holds significant value in the early screening of CP. Additionally, we also focus on the identification of FMs, and the absence of FMs is mainly used to predict early CP³. Therefore, we retained CP related words and used the description of "early screening for CP".

1. Bosanquet M, Copeland L, Ware R, Boyd R. A systematic review of tests to predict cerebral palsy in young children. *Dev Med Child Neurol* 55, 418-426 (2013).
2. Hadders-Algra M. Early diagnosis and early intervention in cerebral palsy. *Front Neurol* 5, 185 (2014).
3. Einspieler C, Prechtl HFR. Prechtl's assessment of general movements: A diagnostic tool for the functional assessment of the young nervous system. *Ment Retard Dev Disabil Res Rev* 11, 61-67 (2005).

- It is unclear from the manuscript how the consistency between predictions of MAM and GMA experts (presented in Figure 4B) was calculated. Please include details on what videos were used (e.g., the external validation dataset) and whether or not the GMA experts annotated FMs in all 10 second clips of these videos.

The concordance between MAM and GMA experts on FMs annotations in the revised manuscript is calculated by Cohen's kappa statistic (ranging from -1 to 1). We have described this in the revised manuscript in L318-323 (L233-237 of the clean version): "As shown in Fig. 2b(left), the median kappa value achieved 0.601 (Q1-Q3 0.357-0.845). The Closeness loss function and the Ref Branch were effective in enhancing MAM's ability to identify FMs. Without using the Closeness loss function, the concordance, shown in Fig.2b (middle), reached a kappa value of 0.224 (Q1-Q3 0.025-0.530). Without using the Closeness loss function and Ref Branch, the concordance dropped to a kappa value of 0.064 (Q1-Q3 -0.046-0.280), as shown in Fig. 2Figure 2b (right)." and L619-621 (L475-477 of the clean version): "Cohen's kappa statistic was used in measuring the reliability between the two GMA experts, and the concordance between MAM and GMA experts on FMs annotations."

The video requirements are firstly mentioned in L68-71 (L43-45 of the clean version): "Health professionals trained in GMA possess the ability to qualitatively assess FMs by observing infants in a supine position, free from crying or external stimuli." We also added statistical description for the videos in L184-193 (L135-143 of the clean version): "Within Cohort 1, the median video duration is 295 seconds, with a range of 119 to 653 seconds. Frame rates predominantly include 25 fps (95.4%), with a smaller proportion at 29 fps (4.6%). Pixel resolutions encompass 720×576 (1.7%), 1280×720 (4.9%), 1440×1080 (1.3%), and 1920×1080 (92.2%). In Cohort 2, the median video duration is 297 seconds, with a range of 181 to 556 seconds. Frame rates are all 25 fps (100%),

and pixel resolutions encompass 1280×720 (10.4%), 1440×1080 (5.4%), and 1920×1080 (89.6%). Cohort 3 exhibits a median video duration of 299 seconds, ranging from 188 to 409 seconds. Frame rates in this cohort are predominantly 25 fps (96.3%), with a smaller proportion at 29 fps (3.7%). Pixel resolutions encompass 720×576 (2.9%), 1280×720 (16.9%), 1440×1080 (2.1%), and 1920×1080 (78.2%).” Figure 1 also includes video descriptions (“No supine position in the video”, “Crying or unawake in the video”, and “Video duration < 1 minute”) that do not conform to GMA recording standards:

Figure 1. The flowchart of cohort filtering.

a Cohort 1 initially comprised 1204 infants. 298 infants were excluded, and 906 infants were retained for training and internal cross-validation, including 691 infants categorized as having normal FMs and 215 infants categorized as having risk FMs (absent FMs or sporadic FMs). **b** Cohort 2 initially comprised 283 infants. 62 infants were excluded, and 221 infants were retained for external validation, including 173 infants categorized as having normal FMs and 48 infants categorized as having risk FMs. The infants in normal group included 60 infants having continuous FMs and 113 infants having intermittent FMs. **c** Cohort 3 is the 298 infants excluded from Cohort 1. 55 infants were excluded due to age, video quality or abnormal FMs, and 243 infants were retained for MAM pre-training, whose videos were annotated with FMs and further segmented into 1,586 FMs clips and 4,100 non-FMs clips. FMs fidgety movements.

It is difficult for GMA experts to annotate FMs in a short period of time or frame by frame, as they need to repeatedly watch the entire video and make interval judgments about the FMs. Cohort 3 was annotated for MAM pre-training and serves as reference, and Cohort 2 was annotated for assessing concordance between MAM and GMA experts. Cohort 1 was not annotated. We added the description of how we obtained a 9.6 second annotation as shown in L477-483 (L356-362 of the clean version): “In Cohort 3, all videos were annotated for the intervals of FMs appearance. We cropped 1586 FMs clips that are completely covered by FMs and 4100 non-FMs clips that contain

no FMs, each 9.6 seconds long. All the videos in Cohort 2 were also annotated for the intervals of FMs appearance. These videos were further segmented into 9.6-second clips with a step size of 6 seconds. Based on whether the proportion of FMs within each clip was greater than 0.5, these clips were categorized as FMs clips or non-FMs clips. This categorization was used to assess the concordance between MAM and GMA experts in identifying FMs.”

- Provide statistics on the video material, include range and median values for video length, spatial resolution and temporal resolution for each of the three cohorts.

We added the statistics in L184-193 (L135-143 of the clean version): “Within Cohort 1, the median video duration is 295 seconds, with a range of 119 to 653 seconds. Frame rates predominantly include 25 fps (95.4%), with a smaller proportion at 29 fps (4.6%). Pixel resolutions encompass 720×576 (1.7%), 1280×720 (4.9%), 1440×1080 (1.3%), and 1920×1080 (92.2%). In Cohort 2, the median video duration is 297 seconds, with a range of 181 to 556 seconds. Frame rates are all 25 fps (100%), and pixel resolutions encompass 1280×720 (10.4%), 1440×1080 (5.4%), and 1920×1080 (89.6%). Cohort 3 exhibits a median video duration of 299 seconds, ranging from 188 to 409 seconds. Frame rates in this cohort are predominantly 25 fps (96.3%), with a smaller proportion at 29 fps (3.7%). Pixel resolutions encompass 720×576 (2.9%), 1280×720 (16.9%), 1440×1080 (2.1%), and 1920×1080 (78.2%).”

- In “3D Information Acquisition”: describe the specific post-processing techniques used for smoothing and normalizing coordinate data along with their hyperparameters (e.g., window size of filter).

We added the specific post-processing techniques in L495-497 (L373-375 of the clean version): “Moving Average process was applied to the obtained coordinate values with a window size of 5 frames. We then calculated the mean and standard deviation of all coordinates in each of the three dimensions and used this information to normalize the coordinates.”

- In “Construction details and loss functions of MAM”: mention which distance function are used by the loss functions.

We added the related descriptions in L531 (L409 of the clean version): “ D is the Euclidean distance function” and L548 (L426 of the clean version): “ D is the Euclidean distance function”.

- Include more details on the Transformer models: how many layers in the Spatio Transformer and the Temporal Transformer, what are their embedding dimensions, how many attention heads etc.?

We added the related descriptions in L570-572 (L447-449 of the clean version): “The spatial Transformer utilizes 2 layers of Transformer encoder. The embedding dimension is 64, the number of attention heads is 4, and the number of feedforward network dimension is 256.” and L576-577 (L453-454 of the clean version): “The structure of the Temporal transformer is identical to that of the spatial Transformer.”

- I would recommend to present the main results for the predictive accuracy of MAM and state-of-the-art methods (Figure 3 in current version) in a table where positive and negative predictive values are included in addition to the existing values for AUC, accuracy, sensitivity and specificity.

Thanks for your advice. We have replaced the original Figure 3 with Table 2 as shown below. The positive predictive value and negative predictive value are also included.

Method	AUC	Accuracy	Sensitivity	Specificity	PPV	NPV
Internal cross-validation						
EML	0.846 ± 0.034	0.787 ± 0.017	0.786 ± 0.078	0.788 ± 0.039	0.556 ± 0.033	0.918 ± 0.026
STAM	0.880 ± 0.005	0.840 ± 0.020	0.832 ± 0.030	0.842 ± 0.036	0.643 ± 0.048	0.938 ± 0.008
WO-GMA	0.912 ± 0.010	0.856 ± 0.017	0.879 ± 0.042	0.848 ± 0.025	0.663 ± 0.034	0.955 ± 0.015
MAM	0.973 ± 0.007	0.938 ± 0.007	0.939 ± 0.021	0.934 ± 0.014	0.826 ± 0.031	0.980 ± 0.006
MAM.w/o.info	0.965 ± 0.006	0.931 ± 0.012	0.912 ± 0.010	0.937 ± 0.016	0.832 ± 0.034	0.969 ± 0.003
External validation						
EML	0.844 ± 0.026	0.767 ± 0.027	0.850 ± 0.035	0.744 ± 0.039	0.483 ± 0.037	0.947 ± 0.010
STAM	0.882 ± 0.011	0.810 ± 0.016	0.879 ± 0.018	0.791 ± 0.025	0.539 ± 0.025	0.959 ± 0.004
WO-GMA	0.906 ± 0.014	0.848 ± 0.019	0.904 ± 0.048	0.832 ± 0.035	0.603 ± 0.040	0.970 ± 0.013
MAM	0.967 ± 0.005	0.934 ± 0.008	0.925 ± 0.024	0.936 ± 0.009	0.802 ± 0.022	0.978 ± 0.008
MAM.w/o.info	0.966 ± 0.006	0.928 ± 0.008	0.908 ± 0.038	0.933 ± 0.005	0.790 ± 0.013	0.974 ± 0.011

Table 1. The performance of MAM in internal cross-validation and external validation

Data are represented by mean ± sd. The highest value for each metric is shown in bold. *EML* ensemble machine learning model by Mccay et al.³¹, *STAM* spatio-temporal attention-based model by Nguyen-Thai et al.³², *WO-GMA* weakly supervised online action detection model by Luo et al.³³, *MAM* motor assessment model, *MAM.w/o.info* MAM without the Info Branch. *AUC* area under receiver operating characteristic curve, *PPV* positive predictive value, *NPV* negative predictive value.

- The current version of the manuscript requires polishing as it appears unorganized and unfinished:

We apologize for any inappropriate description, and have made corresponding modifications.

o Methodological descriptions in the Results section (e.g., L105-128: “Construction of a multi-instance multimodal motor assessment model, MAM”, L160-162: “We also applied the SHapley Additive exPlanations...”)

We have changed the subheading “Construction of a multi-instance multimodal motor assessment model, MAM” to “Overall architecture of MAM”, and changed the way we describe this section in L202-237 (L152-174 of the clean version): “MAM represents a multi-instance multimodal model founded on the Transformer architecture. As depicted in Fig. 2, MAM is organized into three key components: the Ref Branch, the Main Branch, and the Info Branch. The Ref Branch is specifically

designed for pre-training the spatio-temporal Transformer (Supplementary Fig. 1) and the classifier within MAM. It utilizes video clips featuring FMs and non-FMs from Cohort 3 (as detailed in Fig. 1). The training process incorporates the Triplet loss⁴¹, a form of margin loss, to distinguish between FMs clip representations and non-FMs clip representations. These representations are dynamically adapted during training and serve as crucial references for the clip representations in the Main Branch. Both the Main Branch and Ref Branch share a substantial portion of their architectural elements. The fine-tuned 3D pose estimation model (see Methods) is responsible for extracting critical joint coordinates from supine infant videos. Importantly, the input construction step involves distance matrices of each dimension (see Methods), which encapsulate overall motion patterns and capture potential coordination relationships among non-adjacent joints. The Main Branch incorporates an additional split step, dividing the input (referred to as a “bag” in MIL) into small parts (“instance” in MIL) of the same size as those in the Ref Branch. Meanwhile, an extra attention-based fusion (Supplementary Fig. 2) step is responsible for amalgamating the probabilities of FMs and non-FMs instances into the normal and risk probability of the “bag”. The designed Closeness loss function (see Methods) enables instances’ clip representations to dynamically converge towards the FMs or non-FMs representation centers of the Ref Branch, resulting in improved instance prediction outcomes. The Info Branch utilizes the basic characteristics (sex, gestational age, birth weight, and assessment age) of the infants corresponding to the videos in the Main Branch for predictions. The resultant predictions are integrated with those from the Main Branch to yield multimodal predictions for the final normal and risk probability.”

The description of the use of SHAP was changed to “Furthermore, SHapley Additive exPlanations (SHAP)⁴² method was applied to explore the contribution of the infants’ characteristics in the Info Branch to the final prediction. As shown in Supplementary Fig. 5, we found that the video feature in the Main Branch dominated the prediction of normal probability, wherein the larger the video feature value after fusion, the greater the prediction probability of the normal group. The contribution degree of the four basic characteristics was small, and it was difficult to distinguish the direction of their influence on the prediction of normal probability.” in L284-290 (L200-206 of the clean version).

o Subjective interpretations under Results (e.g., L141-143: “MAM achieved an extremely high area under the curve... Additionally, MAM’s performance on the external validation dataset was also impressive...”)

We switched to an objective description in L253-261 (L181-189 of the clean version): “The performance of MAM in GMs prediction at the FMs stage was tested both in internal and external datasets. The receiver operating characteristic (ROC) curve of each fold in internal cross-validation and mean ROC curve in external validation are shown in Supplementary Fig. 3 and Supplementary Fig. 4. MAM achieved an area under ROC curve (AUC) value of 0.973 (\pm 0.007), an accuracy of 0.938 (\pm 0.007), a sensitivity of 0.939 (\pm 0.021), a specificity of 0.934 (\pm 0.014), a positive predictive value (PPV) of 0.826 (\pm 0.031) and a negative predictive value (NPV) of 0.980 (\pm 0.006) in the

internal dataset. In the external validation, MAM achieved an AUC value of 0.967 (\pm 0.005), an accuracy of 0.934 (\pm 0.008), a sensitivity of 0.925 (\pm 0.024), a specificity of 0.936 (\pm 0.009), a PPV of 0.802 (\pm 0.022) and a NPV of 0.978 (\pm 0.008).”

o Unreferenced claims (L70-71: “Although artificial intelligence-based automatic methods demonstrate high accuracy in predicting CP, they still fall short in achieving quantitative prediction.”)

We revised this paragraph and included relevant references in L104-L124 (L68-L81 of the clean version): “In the context of automated GMA methods at the FMs stage, high accuracy is typically achieved through qualitative approaches that leverage spatiotemporal models. However, these methods often provide only the final classification outcome for GMs^{32, 33}. In contrast, quantitative tools offer objective measurements and numerical data, reducing the influence of subjective interpretation and enabling a more comprehensive assessment of a patient's condition^{35, 36}. Some studies have attempted to quantify GMA by analyzing the motion patterns in various body parts, including the head, trunk, arms, and legs^{30, 31}. These analyses consider factors such as direction, magnitude, speed, and acceleration, along with their statistical characteristics, to derive quantitative values^{26, 27, 30, 31}. Nevertheless, quantitative approaches have not consistently matched the performance of qualitative methods in GMs classification, and the emphasis on different body parts or various indicators contradicts the spatial integrity in gestalt perception³⁷. Moreover, both qualitative and quantitative models often lack the interpretability required for health professionals to fully comprehend the results. While some studies have acknowledged the occurrence of FMs, their efforts to explain the precise locations where FMs appear have been insufficient^{32, 33}.”

26. Adde L, Helbostad JL, Jensenius AR, Taraldsen G, Støen R. Using computer-based video analysis in the study of fidgety movements. *Early Hum Dev* 85, 541-547 (2009).

27. Støen R, et al. Computer-based video analysis identifies infants with absence of fidgety movements. *Pediatr Res* 82, 665-670 (2017).

30. McCay KD, Ho ESL, Shum HPH, Fehringer G, Marcroft C, Embleton ND. Abnormal Infant Movements Classification With Deep Learning on Pose-Based Features. *IEEE Access* 8, 51582-51592 (2020).

31. McCay KD, et al. A Pose-Based Feature Fusion and Classification Framework for the Early Prediction of Cerebral Palsy in Infants. *IEEE Trans Neural Syst Rehabil Eng* 30, 8-19 (2022).

32. Nguyen-Thai B, Le V, Morgan C, Badawi N, Tran T, Venkatesh S. A Spatio-Temporal Attention-Based Model for Infant Movement Assessment From Videos. *IEEE J Biomed Health Inform* 25, 3911-3920 (2021).

33. Luo T, et al. Weakly Supervised Online Action Detection for Infant General Movements. In: *Medical Image Computing and Computer Assisted Intervention – MICCAI 2022* (eds Wang L, Dou Q, Fletcher PT, Speidel S, Li S). Springer Nature Switzerland (2022).

35. Mordini FE, et al. Diagnostic accuracy of stress perfusion CMR in comparison with quantitative coronary angiography: fully quantitative, semiquantitative, and qualitative assessment. *JACC Cardiovasc Imaging* 7, 14-22 (2014).

36. Lakshman M, Sinha L, Biswas M, Charles M, Arora N. Quantitative vs qualitative research methods. *The Indian Journal of Pediatrics* 67, 369-377 (2000).

37. Koffka K. Perception: An introduction to the Gestalt-theorie. Psychological bulletin 19, 531-585 (1922).

o Results that are not part of the Results section in the manuscript receive much attention in the beginning of the Discussion section (e.g., L233-239: “Further, we evaluated the AUC values obtained by the model using different input construction ways and pose estimation methods of different dimensions (Table S2) ...”)

We deleted this part in the Discussion section and wrote the related contents in the Results section in L291-296 (L207-212 of the clean version): “To illustrate the role of distance matrices and 3D pose estimation, we evaluated the AUC values obtained by the model using different input construction ways and pose estimation methods of different dimensions (Supplementary Table 2). We found that the proposed input construction way of distance matrices outperformed that of using coordinates, velocities and accelerations, or their combinations. In addition, under the same input construction way, the performance obtained by using 3D pose estimation method is superior to that obtained by using 2D pose estimation method.”

o Incomplete figures (e.g., Figure 3: lacking legend for subpanel D and I would also suggest including labels in the subpanels indicating whether the results are from cross-validation or external validation, Figure S3: legend indicates that standard deviation is presented).

Thanks for your careful review and suggestions. The Figure 3 has been replaced by Table 2 with words indicating cross-validation or external validation. The legend of the original Figure S3 was wrong and the new changes are as follows:

Supplementary Fig. 3. The receiver operating characteristic (ROC) curve of each fold in internal cross-validation.

Each line represents one fold in 5-fold cross-validation. *AUC* area under the ROC curve.

Reply to Reviewer # 2

This paper is reporting on a deep-learning based method for classification of General Movements, which is part of a screening process for CP. GMs assessment is the most reliable method to predict early cerebral palsy, but it is labour intensive as it is dependent on a trained individual scoring the video and it is not practical at a general population level. This is important as 50% of babies who go on to be diagnosed with CP have had an uneventful perinatal period and have gone home without admission to a Neonatal intensive Care unit or special care nursery.

There are several groups around the world investigating this and the authors report high levels of accuracy compared to previous studies. It's important throughout the paper to highlight that this AI model is looking at GMA which is part of the CP screening process and results of this study will be further validated when the outcome of CP is known. It is possible that the children with absent fidgety movements have other neurodevelopmental impairments other than CP. Given the cohort was recruited between 2013-2019, it would be useful to include information about outcomes.

We greatly appreciate your valuable comments and suggestions. We have made some changes to highlight that the AI model aims to automate GMA at the FMs stage. We changed the title from “Deep learning-based quantitative motor assessment for early diagnosis of cerebral palsy” to “Automating General Movements Assessment with quantitative deep learning to facilitate early screening of cerebral palsy”. And we revised the Abstract to: “The Prechtl General Movements Assessment (GMA) is increasingly recognized for its role in evaluating the integrity of the developing nervous system and predicting motor dysfunctions, particularly in conditions such as cerebral palsy (CP). However, the necessity for highly trained professionals has hindered the widespread adoption of GMA as an early screening tool. In this study, we propose a deep learning-based motor assessment model (MAM) that combines infant videos and basic characteristics, with the aim of automating GMA at the fidgety movements (FMs) stage. MAM demonstrates strong performance, achieving an Area Under the Curve (AUC) of 0.967 during external validation. Importantly, it adheres closely to the principles of GMA and exhibits robust interpretability, as it can accurately identify FMs within videos, showing substantial agreement with expert assessments. Leveraging the predicted FMs frequency, a quantitative GMA method is introduced, which achieves an AUC of 0.956 and enhances the diagnostic accuracy of GMA beginners by 11.0%. The development of MAM holds the potential to significantly streamline early CP screening and revolutionize the field of video-based quantitative medical diagnostics.” The introduction to GMA has also been moved to the beginning of the article: “The Prechtl General Movements Assessment (GMA) serves as a valuable tool for assessing the developmental status of an infant's nervous system and the potential presence of motor abnormalities¹. It accomplishes this by evaluating the quality of general movements (GMs), which are part of the infant's repertoire of spontaneous movements^{1, 2}. Typically, when infants reach a corrected age of 9-20 weeks, GMs exhibit characteristics such as

moderate speed, variable acceleration, and engagement of the neck, trunk, and limbs in various directions^{1,2,3}. These specific movements are referred to as fidgety movements (FMs).”

Because the purpose of this study is not to directly assess the accuracy of AI models in CP prediction, we did not establish a complete follow-up dataset. Among the 215 at-risk infants (absent FMs and sporadic FMs) in Cohort 1, we found 156 retained their CP diagnostic results at ages 2-4. We have included this information in the discussion section in L428-432 (L317-321 of the clean version): “Among the risk group in Cohort 1, 156 infants retained their follow-up results at 2-4 years of age, of which 129 (82.7%) were eventually diagnosed with CP. The majority of these infants have received timely intervention after GMA, so the proportion of infants who were diagnosed as only having mild CP was 88.4%. This highlights the significance of GMA in early screening and enabling early intervention.”

Summary

1. Cerebral Palsy is not usually referred to as a syndrome. Central dyskinesia is not usually in the definition.

We apologize for these inappropriate descriptions. These sentences have been deleted, and we now attempt to highlight GMA at the beginning of the abstract: “The Prechtl General Movements Assessment (GMA) is increasingly recognized for its role in evaluating the integrity of the developing nervous system and predicting motor dysfunctions, particularly in conditions such as cerebral palsy (CP). However, the necessity for highly trained professionals has hindered the widespread adoption of GMA as an early screening tool. In this study, we propose a deep learning-based motor assessment model (MAM) that combines infant videos and basic characteristics, with the aim of automating GMA at the fidgety movements (FMs) stage. MAM demonstrates strong performance, achieving an Area Under the Curve (AUC) of 0.967 during external validation. Importantly, it adheres closely to the principles of GMA and exhibits robust interpretability, as it can accurately identify FMs within videos, showing substantial agreement with expert assessments. Leveraging the predicted FMs frequency, a quantitative GMA method is introduced, which achieves an AUC of 0.956 and enhances the diagnostic accuracy of GMA beginners by 11.0%. The development of MAM holds the potential to significantly streamline early CP screening and revolutionize the field of video-based quantitative medical diagnostics.”

Introduction

1. Justification around need for AI would be better supported by the need for scaling (page 3, line 49-50) rather than beginners. The method has good reliability when using the Prechtl method and doing training.

Thanks for your suggestions. We changed the sentences to “While GMA has demonstrated commendable performance, it remains labor-intensive, time-consuming, and requires years of experience, coupled with regular calibration, to achieve the desired precision and consistency^{4, 15}.”

Consequently, the widespread use of GMA in the general population is not feasible, necessitating the exploration of alternative automated GMA methods.” in L75-80 (L46-50 of the clean version)

Results

1. Sporadic fidgety is usually classified as absent fidgety. There are 12 infants with sporadic fidgety in the FM+ category. This will affect the judgement and effectiveness of the tool.

Thanks for your constructive suggestions. We have grouped "sporadic FMs" and "absent FMs" into the same category, as described in L165-169 (L118-122 of the clean version): “It is worth noting that normal FMs can be further subdivided into continuous FMs, intermittent FMs, and sporadic FMs. The prognosis associated with sporadic FMs closely resembles that of absent FMs^{39, 40}. Consequently, we have categorized infants with continuous FMs and intermittent FMs as the normal group, and infants with sporadic FMs and absent FMs as the risk group.” The results of the entire manuscript were adjusted accordingly.

2. It was difficult to be sure that the tool could differentiate between intermittent and sporadic fidgety movements.

Under the new categorization scheme, as described in our last response, the retrained model effectively distinguished between the normal group and the risk group, as well as between continuous FMs and intermittent FMs based on the FMs frequency it identified. The corresponding results are described in L324-344 (L238-253 of the clean version):

Figure 2. The performance of MAM in FMs identification and quantification.

a Examples of FMs and non-FMs clips evaluated by MAM and GMA experts. Each square represents the predicted result of a 9.6-second-long clip of the video, with blue denoting a non-FMs clip and pink denoting an FMs clip. **b** Distribution of concordance in the evaluations of FMs in videos from the external validation dataset by GMA experts and different MAM types. MAM.w/o.C MAM without Closeness loss function, MAM.w/o.C.ref MAM without Closeness loss function and Ref Branch. **c** Distribution of predicted FMs frequencies by MAM in normal group and risk group. **d** The ROC curve for classification of normal and risk group using predicted FMs frequency by MAM. **e** Distribution of predicted FMs frequencies by MAM in infants having continual FMs and intermittent FMs.

We calculated the proportion of FMs clips in an entire video and named it “FMs frequency”. As shown in Fig. 2c, the median of FMs frequency in the external validation dataset is 0.553 (Q1-Q3 0.412-0.706) in the normal group and 0.135 (Q1-Q3 0.082-0.215) in the risk group. There is a significant difference ($p < 0.01$) in FMs frequency between the normal group and the risk group.

Further, we classified videos of the normal group and risk group directly using the FMs frequency to explore if MAM could transform GMA from a qualitative to a quantitative tool. As shown in Fig. 2d, an AUC of 0.956 (95% CI 0.924-0.989) is obtained, indicating that we can quantify GMA in this way.

We then observed the distribution of FMs frequency evaluated by MAM in infants having continuous FMs and intermittent FMs. As shown in Fig. 2e, the median FMs frequency in infants having continuous FMs is 0.738 (Q1-Q3 0.672-0.798), and the median FMs frequency in infants having intermittent FMs is 0.455 (Q1-Q3 0.359-0.560). There is a significant difference ($p < 0.01$) in FMs frequency between these two categories. Using a threshold of 0.603 to classify continuous FMs and intermittent FMs based on FMs frequency, results in an AUC of 0.932 (95% CI 0.892-0.973) and an accuracy of 0.902. These results further validate that our approach can quantitatively automate GMA.

Discussion

1. The results should be presented in relation to the GMA rather than CP, as you don't yet know how many of these children went on to have CP.

We apologize for these inappropriate descriptions. The “CP” in the corresponding sentences have been changed to “GMs” related words. L374-377 (L282-284 of the clean version): “In this study, we developed a multi-instance multimodal motor assessment model based on the Transformer architecture to realize prediction of GMs in infants at the FMs stage, which aimed at expediting the early detection of CP.”

2. Through the manuscript the authors refer to “doctors” using GMA but it is used by a range of health professionals around the world and I would recommend “doctor” be replaced with “health professional”

Thanks for your professional suggestions. We have replaced “doctor” with “health professional” throughout the manuscript. L68-72 (L43-45 of the clean version): “Health professionals trained in GMA possess the ability to qualitatively assess FMs by observing infants in a supine position, free from crying or external stimuli. They categorize an infant's GMs at the FMs stage as normal FMs, abnormal FMs, or absent FMs^{1, 2, 13, 14}.” L360-L362 (L268-270 of the clean version): “In clinical practice, the health professionals need to distinguish FMs from various other movements that occur concurrently with FMs, such as wiggling-oscillating and saccadic arm movements^{2, 14}.” L398-402 (L292-296 of the clean version): “The perception of FMs is the prerequisite for health professionals to make correct GMs classification judgment at the FMs stage, and the absence of FMs constitutes the most robust indicator for predicting CP^{4, 40}. Therefore, if the model can pinpoint the location of FMs within videos, it not only provides decision support for healthcare professionals but also enhances the model's transparency and reliability.” L413-414 (L303-304 of the clean version): “Quantification has always been a focus in medical diagnostic tools, as it provides more objective,

intuitive, and convenient evidence for both health professionals and patients^{35, 36.}”

3. On page 15, the authors write that Early diagnosis of CP is of the utmost importance. It would be important to explain the reasons for this which include supporting the suspicions of parents, being able to reassure them about outcomes and provide targeted early intervention to improve long term outcomes. This will also allow early screening for other co-morbidities such as visual and hearing impairments, enable active surveillance of hips and prevention of contractures and scoliosis.

Thanks for your professional suggestions. Here is our additional explanation of why early diagnosis of CP is important, using your suggestions and the reasoning of other papers. L424-428 (L313-317 of the clean version): “Early screening of CP ensures that the interventions can be implemented at critical stages of brain development to maximize motor and cognitive outcomes, and reduce the incidence of other co-morbidities such as visual and hearing impairments^{6, 9}. Additionally, early screening provides parents with a clear result for their high-risk baby, reducing their stress and anxiety, and enhancing their coping abilities^{9, 44.}”

6. Hadders-Algra M. Early diagnosis and early intervention in cerebral palsy. *Front Neurol* 5, 185 (2014).

9. Morgan C, et al. Early Intervention for Children Aged 0 to 2 Years With or at High Risk of Cerebral Palsy International Clinical Practice Guideline Based on Systematic Reviews. *JAMA Pediatr* 175, 846-858 (2021).

44. Novak I, et al. Early, Accurate Diagnosis and Early Intervention in Cerebral Palsy Advances in Diagnosis and Treatment. *JAMA Pediatr* 171, 897-907 (2017).

Methods

1. Statistical analysis review is not included in this review. The journal may require separate advice on the statistical methodology and analysis and the mathematical equations from pages 15 to 19.

	Internal cross-validation dataset (n = 906)			External validation dataset (n = 221)		
	Normal group	Risk group	p value	Normal group	Risk group	p value
n	691 (76.3%)	215 (23.7%)		173 (78.3%)	48 (21.7%)	
Sex						
male	353 (51.1%)	114 (53.0%)	0.676	87 (50.0%)	27 (56.3%)	0.570
female	338 (48.9%)	101 (47.0%)		86 (50.0%)	21 (43.8%)	
GA (week)	35.11 (3.22)	35.68 (3.22)	0.012	35.14 (3.34)	35.58 (3.38)	0.384
BW (g)	2,415.67 (665.88)	2,514.93 (699.11)	0.093	2,414.86 (687.62)	2,526.56 (671.03)	0.313
CA (week)	12.53 (2.72)	12.53 (2.64)	0.866	12.73 (2.72)	12.75 (2.86)	0.928

Table 2. The basic characteristics of the infants in internal and external datasets.

Data are represented by *n* (%) or mean (sd). The *p* values are calculated using Chi-square test, t-test or Mann-Whitney test. All tests are two-tailed. *GA* gestational age, *BW* birth weight, *CA* corrected age.

We gave statistical analysis about the infants’ basic characteristics and videos in L172-180 (L124-

130 of the clean version): “Comprehensive details regarding the basic characteristics of the infants included in both the internal and external datasets are provided in Table Table 2. To assess potential differences between the normal and risk groups, we conducted a Chi-square test to examine sex distribution, while other essential characteristics, such as gestational age, birth weight, and corrected age, were analyzed using either the t-test or Mann-Whitney test, depending on the normality of the data. Notably, no statistically significant differences ($p > 0.01$ for all comparisons) were observed between these two groups.” and L184-193 (L135-143 of the clean version): “Within Cohort 1, the median video duration is 295 seconds, with a range of 119 to 653 seconds. Frame rates predominantly include 25 fps (95.4%), with a smaller proportion at 29 fps (4.6%). Pixel resolutions encompass 720×576 (1.7%), 1280×720 (4.9%), 1440×1080 (1.3%), and 1920×1080 (92.2%). In Cohort 2, the median video duration is 297 seconds, with a range of 181 to 556 seconds. Frame rates are all 25 fps (100%), and pixel resolutions encompass 1280×720 (10.4%), 1440×1080 (5.4%), and 1920×1080 (89.6%). Cohort 3 exhibits a median video duration of 299 seconds, ranging from 188 to 409 seconds. Frame rates in this cohort are predominantly 25 fps (96.3%), with a smaller proportion at 29 fps (3.7%). Pixel resolutions encompass 720×576 (2.9%), 1280×720 (16.9%), 1440×1080 (2.1%), and 1920×1080 (78.2%).”

Other statistical analysis is described in L611-630 (L472-L486 of the clean version) as follows:

Statistical analysis

All hypothesis tests were two-tailed and used a significance level of 0.01. For categorical variables, the Chi-Square test was employed, while for continuous variables, either a t-test or Mann-Whitney test was chosen based on the fulfillment of normality assumptions. Cohen's kappa statistic was used in measuring the reliability between the two GMA experts, and the concordance between MAM and GMA experts on FMs annotations.

The accuracy, sensitivity, specificity, PPV and NPV were used to measure the model performance. These metrics are defined as follows:

$$\text{accuracy} = \frac{N_{TP} + N_{TN}}{N_{TP} + N_{FP} + N_{TN} + N_{FN}} \quad (10)$$

$$\text{sensitivity} = \frac{N_{TP}}{N_{TP} + N_{FN}} \quad (11)$$

$$\text{specificity} = \frac{N_{TN}}{N_{TN} + N_{FP}} \quad (12)$$

$$\text{PPV} = \frac{N_{TP}}{N_{TP} + N_{FP}} \quad (13)$$

$$\text{NPV} = \frac{N_{TN}}{N_{TN} + N_{FN}} \quad (14)$$

where N_{TP} , N_{TN} , N_{FP} , N_{FN} represent the number of true positive, true negative, false positive, false negative infants, respectively.

2. Further information is required on how the GMA scores were classified by humans. What methods of GMs was used - Prechtl or Hadders-Algra? There are references included to both methods. How did you define a “beginner” vs “GMA expert”. What was the reliability between the two GMA experts?

The GMA assessor just gives a classification result without a score. The assessment way is described in L68-72 (L43-46 of the clean version): “Health professionals trained in GMA possess the ability to qualitatively assess FMs by observing infants in a supine position, free from crying or external stimuli. They categorize an infant’s GMs at the FMs stage as normal FMs, abnormal FMs, or absent FMs^{1, 2, 13, 14}.”

We used the Prechtl GMA, and related descriptions are added in the Abstract: “The Prechtl General Movements Assessment (GMA) is increasingly recognized for its role in evaluating the integrity of the developing nervous system and predicting motor dysfunctions, particularly in conditions such as cerebral palsy (CP).” and at the beginning of Introduction: “The Prechtl General Movements Assessment (GMA) serves as a valuable tool for assessing the developmental status of an infant’s nervous system and the potential presence of motor abnormalities¹.”

The Hadders-Algra’s paper below is used to be a reference for “gestalt perception” in L68-72 (L43-46 of the clean version): “Health professionals trained in GMA possess the ability to qualitatively assess FMs by observing infants in a supine position, free from crying or external stimuli. They categorize an infant’s GMs at the FMs stage as normal FMs, abnormal FMs, or absent FMs^{1, 2, 13, 14}.” as this paper writes “Currently, there are two variants in assessing the quality of GMs: Prechtl’s method and Hadders-Algra’s classification ... Nevertheless, they both qualify the GMs based on the Gestalt perception ...”

13. Wu YC, van Rijssen IM, Buurman MT, Dijkstra LJ, Hamer EG, Hadders-Algra M. Temporal and spatial localisation of general movement complexity and variation-Why Gestalt assessment requires experience. *Acta Paediatr* 110, 290-300 (2021).

We added the description of “GMA expert” in L468-L469 (L350-351 of the clean version): “Both experts held GMA certification and had over five years of assessment experience.” The description for “beginners” is in L367-L369 (L275-277 of the clean version): “Using GMA expert evaluations as reference, we compared the diagnostic accuracy of three GMA beginners (GMA certified for less than six months) with and without the assistance of MAM on the external validation dataset.”

The reliability between the two GMA experts is added in L471 (L352-253 of the clean version): “The reliability between the two experts reached a kappa value of 0.947.”

REVIEWERS' COMMENTS

Reviewer #1 (Remarks to the Author):

Recommend to move "Overall architecture of MAM" to the Methods section but otherwise no further comments.

Reviewer #5 (Remarks to the Author):

The authors have mostly responded to the reviewer comments and the manuscript is improved. However, there are a few minor changes suggested.

1. "However, the necessity for highly trained professionals has hindered the widespread adoption of GMA as an early screening tool."

-I would add "hindered widespread adoption IN SOME COUNTRIES". It is used widely in many places.

2. "While GMA has demonstrated commendable performance, it remains labor-intensive, time-consuming, and requires years of experience, coupled with regular calibration, to achieve the desired precision and consistency."

-I would tone down the language here. It only takes 2 minutes to score the video. Assessors are qualified after a 3.5-day course – it doesn't take years. Your AI model will only save a few minutes for scoring but the videos still need to be acquired and information communicated to families.

3. The Hadders-Algra's paper below is used to be a reference for "gestalt perception" in L68-72 (L43- 46 of the clean version): "Health professionals trained in GMA possess the ability to qualitatively assess FMs by observing infants in a supine position, free from crying or external stimuli. They categorize an infant's GMs at the FMs stage as normal FMs, abnormal FMs, or absent FMs^{1, 2, 13, 14.}" as this paper writes "Currently, there are two variants in assessing the quality of GMs: Prechtl's method and Hadders-Algra's classification ... Nevertheless, they both qualify the GMs based on the Gestalt perception ..."

-I would suggest you use a reference to the Prechtl method and gestalt rather than Hadders-Algra to not confuse the reader about the two methods

Response to reviewers

Manuscript ID: NCOMMS-23-28586A

Title: Deep learning-based quantitative motor assessment for early diagnosis of cerebral palsy

New title: Automating General Movements Assessment with quantitative deep learning to facilitate early screening of cerebral palsy

We once again appreciate the reviewers sincerely for their positive feedback and constructive suggestions on our manuscript. The following are a copy of the comments (marked in blue) and our detailed point-by-point response. The manuscript content modified in response to the suggestions is indicated in orange in this file and is also highlighted in the revised manuscript.

Reply to Reviewer # 1

Recommend to move "Overall architecture of MAM" to the Methods section but otherwise no further comments.

We greatly appreciate your valuable suggestions. We have moved this part to the Methods section and made some adjustments to the corresponding changes.

Reply to Reviewer # 5

The authors have mostly responded to the reviewer comments and the manuscript is improved. However, there are a few minor changes suggested.

We greatly appreciate your valuable comments and suggestions. We have made the corresponding modifications according to your suggestions.

1. "However, the necessity for highly trained professionals has hindered the widespread adoption of GMA as an early screening tool."

-I would add "hindered widespread adoption IN SOME COUNTRIES". It is used widely in many places.

Thanks for your suggestions. We have changed the sentence to: "However, the necessity for highly trained professionals has hindered the adoption of GMA as an early screening tool in some countries."

2. "While GMA has demonstrated commendable performance, it remains labor-intensive, time-consuming, and requires years of experience, coupled with regular calibration, to achieve the desired precision and consistency."

-I would tone down the language here. It only takes 2 minutes to score the video. Assessors are qualified after a 3.5-day course – it doesn't take years. Your AI model will only save a few minutes for scoring but the videos still need to be acquired and information communicated to families.

Thanks for your suggestions. We have changed the sentence to: "... in which the absent FMs demonstrate a sensitivity of 98% and a specificity of 94% in CP prediction. Nevertheless, GMA still requires highly trained and certified professionals, as well as experience and regular calibration, to achieve the desired precision and consistency."

3. The Hadders-Algra's paper below is used to be a reference for "gestalt perception" in L68-72 (L43- 46 of the clean version): "Health professionals trained in GMA possess the ability to qualitatively assess FMs by observing infants in a supine position, free from crying or external stimuli. They categorize an infant's GMs at the FMs stage as normal FMs, abnormal FMs, or absent FMs1, 2, 13, 14." as this paper writes "Currently, there are two variants in assessing the quality of GMs: Prechtl's method and Hadders-Algra's classification ... Nevertheless, they both qualify the GMs based on the Gestalt perception ..."

-I would suggest you use a reference to the Prechtl method and gestalt rather than Hadders-Algra to not confuse the reader about the two methods

Thanks for your thoughtful suggestions. This reference has been removed and the references are now: "Health professionals trained in GMA possess the ability to qualitatively assess FMs by

observing infants in a supine position, free from crying or external stimuli. They categorize an infant's GMs at the FMs stage as normal FMs, abnormal FMs, or absent FMs^{1, 2, 11, ...}”

1. Prechtl HFR, Einspieler C, Cioni G, Bos AF, Ferrari F, Sontheimer D. An early marker for neurological deficits after perinatal brain lesions. *Lancet* **349**, 1361-1363 (1997).
2. Einspieler C, Prechtl HFR. Prechtl's assessment of general movements: A diagnostic tool for the functional assessment of the young nervous system. *Ment Retard Dev Disabil Res Rev* **11**, 61-67 (2005).
11. Einspieler C, Prechtl HFR, Bos A, Ferrari F, Cioni G. Prechtl's method on the qualitative assessment of general movements in preterm, term and young infants. *Clin Dev Med* **167**, 1-91 (2004).